# BrainStratify: Coarse-to-Fine Disentanglement of Intracranial Recordings for Speech Decoding

## Abstract

Decoding speech directly from neural activity is a central goal in brain-computer interface (BCI) research. In recent years, exciting advances have been made through the growing use of intracranial field potential recordings, such as stereo-ElectroEncephaloGraphy (sEEG) and ElectroCorticoGraphy (ECoG). These neural signals capture rich population-level activity but present key challenges: (i) task-relevant neural signals are sparsely distributed across sEEG electrodes, and (ii) multiple neural components (e.g., tongue & jaw & lips control in vSMC) are often entangled within the task-relevant functional groups in both sEEG and ECoG recordings. To address these challenges, we introduce a unified speech decoding framework enhanced by Coarse-to-Fine disentanglement, BrainStratify, which includes (i) identifying functional groups through spatial-context-guided temporal-spatial modeling, and (ii) disentangling neural components within the target functional group using Decoupled Product Quantization (DPQ). We evaluate BrainStratify on six datasets (including sEEG, (epidural) ECoG, etc.), spanning tasks like vocal production, speech perception, etc. Extensive experiments show that BrainStratify, as a unified framework for decoding speech from intracranial neural signals, significantly outperforms previous decoding methods. Overall, by combining data-driven stratification with neuroscience-inspired modularity, BrainStratify offers a robust and interpretable solution for decoding speech from intracranial recordings. Code and dataset will be publicly available.

## 1 Introduction

Intracranial neural signals refer to the biometric information collected from the brain through invasive recording techniques (e.g., stereo-ElectroEncephaloGraphy (sEEG) and ElectroCorticoGraphy (ECoG)). Their patterns provide rich and high-resolution insights toward understanding the physiological functions of the brain and the mechanism of related diseases, leading to various applications, including speech decoding (Moses et al., 2021; Metzger et al., 2023; Zheng et al., 2025; Chau et al., 2024), motor intention decoding (Natraj et al., 2025; Silversmith et al., 2021), neurological disorders detection (Zhang et al., 2023; Yuan et al., 2023; Li et al., 2025b), among others. Although many studies (Moses et al., 2021; Zheng et al., 2025; Chau et al., 2024) have recently shown promising results in speech decoding (e.g., vocal production and speech perception) based on sEEG and ECoG, significant challenges in modeling intracranial neural signals remain unresolved.

Compared to ECoG, sEEG provides more depth information from specific brain regions, making it particularly attractive in both brain-computer interface (BCI) applications (Chau et al., 2024; Zheng et al., 2025; Mentzelopoulos et al., 2024) and fundamental neuroscience studies (Subramaniam et al., 2024; Norman et al., 2019; Domenech et al., 2020). Despite their potential, sEEG recordings present a unique challenge. In practice, sEEG electrodes are sparsely distributed across the brain, requiring researchers to first identify task-relevant channels before decoding (Wang et al., 2023; Mentzelopoulos et al., 2024; Zheng et al., 2025). For instance, BrainBERT (Wang et al., 2023) adopts a single-channel (SC) strategy, independently evaluating and ranking channels based on decoding performance. In contrast, Du-IN (Zheng et al., 2025) utilizes a multi-channel (MC) approach, analyzing all channels collectively and ranking them based on the learned weight distribution. Both methods require substantial labeled data for supervision, posing significant challenges, as large-scale

labeling in medical experiments is often prohibitively costly or unfeasible. When labeled data are scarce, the selected channels often fail to align with those containing target neural activity (Figure 1).

Another challenge arises from the nature of intracranial recordings themselves. These recordings capture aggregated neural activity from populations of neurons (Chau et al., 2024). While sEEG can enhance spatial resolution through techniques like bi-polar (or Laplacian) re-reference (Li et al., 2018), intracranial neural signals inherently represent a mixture of signals from multiple neural components (e.g., tongue & jaw & lips control in vSMC for vocal production). Without explicit mechanisms to disentangle neural components within specific brain regions, models struggle to extract fine-grained states from intracranial neural signals, leading to reduced decoding performance.

To tackle these issues, we propose a unified framework for decoding speech from intracranial recordings, Brain-Stratify, that comprises two complementary stages: (1) Coarse Disentanglement Learning and (2) Fine Disentanglement Learning. In Coarse-DL, we pre-train a temporal-spatial model with a spatial context task and cluster channels into functional groups (Buzsáki, 2006) based on the sparse inter-channel attention graph, thus facilitating data-efficient channel selection for intracranial sEEG recordings. In Fine-DL, we utilize Decoupled Product Quantization (DPQ) to disentangle neural components (Metzger et al., 2023; Silva et al., 2024) within target groups to enhance the identification of fine-grained neural states, thus facilitating unified representation learning that boosts performance across diverse decoding paradigms (e.g., classification, sequence labeling, regression).

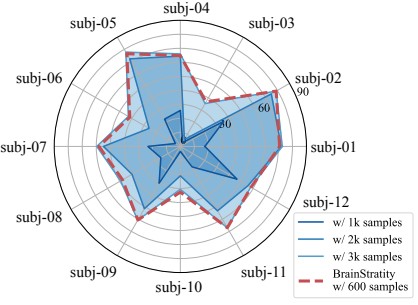

Figure 1: The word classification performance on Du-IN (Zheng et al., 2025) dataset using top-10 channels selected via the MC strategy across varying numbers of labeled samples.

To validate the effectiveness of our proposed framework, we evaluate BrainStratify on six datasets (Appendix B), including sEEG (Zheng et al., 2025; Wang et al., 2024a), (epidural) ECoG, etc. Empirically, Brain-Stratify outperforms existing channel clustering methods (Chen et al., 2025; Qiu et al., 2024), identifying channel cluster that faithfully aligns with those containing target neural activity. Besides, BrainStratify achieves SOTA performance in all decoding tasks, particularly excelling in word decoding (Zheng et al., 2025). This success stems from its ability to effectively integrate modular neural components within the target functional group – a critical requirement for decoding complex, interdependent neural patterns.

To sum up, the main contributions of our work comprise:

1. **Coarse-to-Fine neural disentanglement:** We propose a unified speech decoding framework enhanced by neural disentanglement, BrainStratify, that identifies functional channel groups and disentangles neural components within target groups, via two complementary stages.

2. **Neuroscience-inspired design:** BrainStratify leverages neuroscience insights (e.g., modular brain computation) in its design, discovering task-relevant channel groups based on the sparse inter-channel attention graph learned via self-supervision.

3. **State-of-the-art (SOTA) performance:** Our framework achieves SOTA performance in decoding speech from intracranial neural signals (e.g., sEEG, ECoG) across multiple datasets, demonstrating robust effectiveness across diverse neural recording modalities.

## 2 RELATED WORKS

### 2.1 SELF-SUPERVISED LEARNING IN BCI

Recently, the pre-trained temporal-spatial models (i.e., foundation models) have drawn significant attention across diverse neural modalities, including EEG (Jiang et al., 2024; Wang et al., 2024b;c), sEEG (Wang et al., 2023; Zhang et al., 2023; Chau et al., 2024), fMRI (Caro et al., 2023; Dong et al., 2024), etc. Due to their adaptability in spatial modeling, these models robustly handle varying numbers of channels and excel at channel-level classification tasks (e.g., epilepsy detection). Based on

these models, PopT (Chau et al., 2024) further utilizes `[CLS]` token to aggregate channels, excelling in decoding modular cognitive states (e.g., sentence onset detection) of speech perception.

Other approaches (Zheng et al., 2025; Wu et al., 2024) fuse the manually selected channels into region-level tokens and then pre-train temporal models based on them. While these methods perform well in decoding complex cognitive states (e.g., vocal production), their effectiveness depends on whether the manually selected channels faithfully represent the target functional groups.

## 2.2 CHANNEL CLUSTER IN TIME SERIES

Channel clustering methods, which use cluster information instead of individual channel identities, have gained significant attention in Multivariate Time Series Forecasting (MTSF) research (Chen et al., 2025; Qiu et al., 2024; Huang et al., 2023; Hu et al., 2025). CCM (Chen et al., 2025) employs static cluster embeddings and cross-attention to group channels, improving forecasting performance. Considering its dynamic nature, DUET (Qiu et al., 2024) utilizes correlation-based metric learning to capture the relationship among channels, which are integrated via masked attention for forecasting.

All existing channel cluster methods for time series primarily rely on low-level correlations in raw time series, which are unsuitable for intracranial neural signals. While these signals reflect aggregated neural activity from neuronal populations, similar firing patterns hold distinct functional meanings by location (Buzsáki, 2006), producing inherently multimodal signals across channels. This suggests that correlation-based clustering may inadequately capture functionally relevant neural clusters.

## 2.3 DISENTANGLEMENT REPRESENTATION LEARNING

Disentangling independent latent components from observations is a desirable goal in representational learning (Higgins et al., 2017; Locatello et al., 2019; Wang et al., 2024d), with broad applications in computer vision (Hsu et al., 2023; 2024), time series analysis (Oublal et al., 2024; Woo et al., 2022), and neuroscience (Zhou & Wei, 2020; Wang et al., 2024e; Li et al., 2025a). QLAE (Hsu et al., 2023) leverages learnable latent codex combined with weight decay regularization to extract human-interpretable representations from raw images. Tripod (Hsu et al., 2024) further enhances disentanglement by introducing minimal mixed generator derivatives to guide feature separation.

Unlike images, which typically require disentangling features along [width, height] dimensions, neural signals require disentanglement along the channel dimension – similar to how RGB channels are treated in images. pi-VAE (Zhou & Wei, 2020) leverages supervision to model the relation between latents and task variables. PDisVAE (Li et al., 2025a) encourages group-wise independence in learned representations via partial-correlation constraint to handle non-separable factor entanglement.

## 3 METHOD

The overall framework of BrainStratify is illustrated in Figure 2, where the framework contains two complementary stages: (1) Coarse Disentanglement Learning and (2) Fine Disentanglement Learning (i.e., Disentanglement Representation Learning (DRL)).

## 3.1 TASK DEFINITION

The evaluation spans six datasets (Appendix B), e.g., sEEG, (epidural) ECoG. In Du-IN dataset (Zheng et al., 2025), we assess the word classification (CLS) task, the speech regression (RGS) task, and the syllable sequential classification (CTC) task. In our collected (epidural) ECoG datasets (Appendix A), we evaluate the word & motor CLS task and the syllable CTC task. In Brain Treebank dataset (Wang et al., 2024a), we evaluate four binary-state CLS tasks. In other datasets (Willett et al., 2023; Sivakumar et al., 2024), we evaluate the phoneme & character CTC tasks, respectively.

Multi-channel sEEG signals are represented as $\mathcal{X} \in \mathbb{R}^{C \times T}$, where $C$ is the number of channels and $T$ is the total timestamps. For CLS tasks, the paired label is $y \in \mathcal{Y}$, where $\mathcal{Y}$ represents the label-set. For RGS tasks, the paired label is $y \in \mathbb{R}^{F \times L}$, where $y$ is audio feature (e.g., mel-spectrogram, Wav2vec2 feature (Baevski et al., 2020)), $F$ is the dimension of feature embeddings, and $L$ is the feature timestamps. For CTC tasks, the paired label sequence is $y = \{y_i \in \mathcal{Y} | i = 1, ..., L\}$, where $\mathcal{Y}$ represents the syllable-set and $L$ is the syllable timestamps.

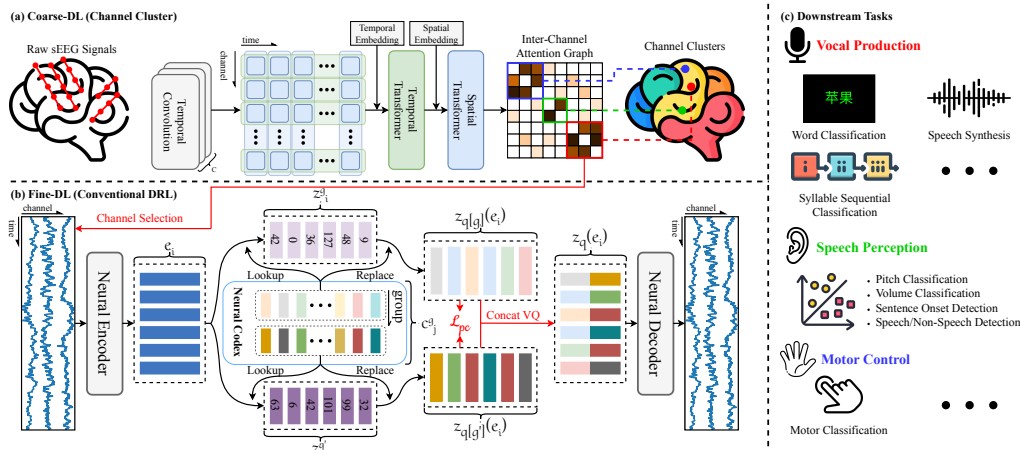

Figure 2: **Overview of BrainStratify framework.** **(a).** Coarse Disentanglement Learning (BrainStratify-Coarse). **(b).** Fine Disentanglement Learning (BrainStratify-Fine). **(c).** Overview of downstream neural decoding tasks.

## 3.2 COARSE DISENTANGLEMENT LEARNING

We introduce BrainStratify-Coarse (Figure 2 (a)), a general architecture for sEEG-based functional group identification. The model contains three parts: (1) Temporal Convolution, (2) Temporal & Spatial Transformer, and (3) Channel Cluster Module. The sEEG signals $\mathcal{X}$ are segmented into patches using a non-overlap $W_c$-length window (0.25s), yielding $\mathcal{X}_p^c = \{\boldsymbol{x}_{i,j}^c \in \mathbb{R}^{W_c} | i = 1, ..., C; j = 1, ..., N_c\}$, where $N_c = \lfloor \frac{T}{W_c} \rfloor$ and the number of patches is $|\mathcal{X}_p^c| = C \times N_c$.

**Temporal Convolution.** Since sEEG signals from different channels are partial observations of whole-brain activity, channel-specific convolution modules are employed to map signals into a unified latent space. Each module encodes patches with stacked 1D convolution layers. The embeddings are $\mathcal{E}_p^c = \{\boldsymbol{e}_{i,j}^c \in \mathbb{R}^d | i = 1, ..., C; j = 1, ..., N_c\}$, where $d$ is the dimension of embeddings.

**Temporal & Spatial Transformer.** We add the parameter-free temporal embeddings introduced in Vaswani et al. (2017) to inject the temporal information. Then, the embeddings are fed into Temporal Transformer to get the temporal-transformed embeddings $\mathcal{E}_t^c$. After that, we add either learnable or MNI-based (Chau et al., 2024) spatial embeddings to inject the spatial information. Finally, the embeddings are fed into Spatial Transformer to get the spatial-transformed embeddings $\mathcal{E}_s^c$.

**Spatial Context Pre-training.** To avoid shortcuts in mask-based reconstruction tasks (Zhang et al., 2023; Wang et al., 2024c), where models might over-rely on intra-channel temporal patterns, we adapt the spatial context task from Chau et al. (2024), refining it for efficient convergence with limited data. Given an sEEG sample $\mathcal{X} \in \mathbb{R}^{C \times T}$, 10% of channels are randomly selected to have their activity replaced by activity from unrelated time points. The model is trained to detect discrepancies between spatial-transformed embeddings $\mathcal{E}_s^c$ and temporal-transformed embeddings $\mathcal{E}_t^c$:

$$\mathcal{L}_c = \text{BCE}(\text{Linear}(||\mathcal{E}_s^c - \mathcal{E}_t^c||_2^2)). \tag{1}$$

With this adaptation, the model converges rapidly to $\geq 95\%$, even with only 1-hour sEEG data.

**Channel Cluster Module.** We compute channel connectivity $\mathcal{P} \in \mathbb{R}^{C \times C}$ by aggregating attention matrices from Spatial Transformer across layers, temporal patches, and sEEG samples. Channels are grouped into functional clusters via spectral clustering (Ng et al., 2001) (Figure 3 (b)); other methods (e.g., hierarchical clustering) yield similar results. Channels within each functional group provide complementary information, collectively encoding a complete and complex neural function. We select and combine these groups based on their performance in downstream decoding tasks.

### 3.3 FINE DISENTANGLEMENT LEARNING

We present BrainStratify-Fine (Figure 2 (b)), a framework for decoding speech from intracranial recordings employing a two-stage pre-training pipeline (VQ-VAE & MAE). To capture fine-grained neural states, we propose Decoupled Product Quantization (DPQ), which disentangles neural components within target functional groups by integrating product quantization (Jegou et al., 2010) with a partial-correlation constraint (Hazarika et al., 2020; Li et al., 2025a) to enforce codex independence.

The model adopts a CNN-Transformer hybrid architecture consisting of (1) Patch Tokenizer and (2) Temporal Transformer. Neural signals are segmented into temporal patches along the temporal dimension. For each sample $\mathcal{X}$, we use a $W_f$-length window (0.15s) with $S_f$-length stride (0.1s), obtaining $\mathcal{X}_p^f = \{\boldsymbol{x}_i^f \in \mathbb{R}^{C \times W_f} | i = 1, ..., N_f\}$, where $N_f = \lfloor \frac{T}{S_f} \rfloor$ is the number of patches.

**Patch Tokenizer.** The patch tokenizer comprises a linear projection and stacked convolution blocks. Each block contains a 1D convolution, group normalization (Wu & He, 2018), and GELU activation (Hendrycks & Gimpel, 2016). The patch embeddings are $\mathcal{E}_p^f = \{\boldsymbol{e}_i^f \in \mathbb{R}^d | i = 1, ..., N_f\}$.

**Temporal Transformer.** We add temporal embeddings to inject temporal information. Then, the embeddings are fed into "Temporal Transformer" to get the temporal-transformed embeddings $\mathcal{E}^f$. For downstream evaluations, we add a randomly initialized MLP to support different tasks.

**Decoupled Product Quantization.** To identify fine-grained functional sub-modules (Metzger et al., 2023; Silva et al., 2024) within target functional groups in VQ-VAE stage, we propose Decoupled Product Quantization (DPQ), using multiple codexes to capture distinct neural components.

The output embeddings $\mathcal{E}^f = \{\boldsymbol{e}_i^f \in \mathbb{R}^d | i = 1, ..., N_f\}$ from "Neural Encoder" are fed into a vector quantizer, which consists of $G$ parallel sub-quantizers (i.e., neural codexes). The $g$-th neural codex is defined as $\mathcal{C}_g = \{\boldsymbol{c}_j^g | j = 1, ..., N_{codex}\} \in \mathbb{R}^{N_{codex} \times d_{codex}}$, where $N_{codex}$ is the number of discrete codes and $d_{codex}$ is the dimension of code embeddings. We utilize a linear projection $\mathbf{z}_{c[g]}$ to get the mapped embeddings $\mathbf{z}_{c[g]}(\mathcal{E}^f) = \{\mathbf{z}_{c[g]}(\boldsymbol{e}_i^f) \in \mathbb{R}^{d_{codex}} | i = 1, ..., N_f\}$ in the codex space. Then, the codex looks up the nearest neighbor of each embedding $\mathbf{z}_{c[g]}(\boldsymbol{e}_i^f)$ in the neural codex $\mathcal{C}_g$.

$$
\begin{aligned}
\mathbf{z}_{q[g]}(\mathcal{E}^f) &= \{\mathbf{z}_{q[g]}(\boldsymbol{e}_i^f) | i = 1, ..., N_f\}, \\
\mathbf{z}_{q[g]}(\boldsymbol{e}_i^f) &= \boldsymbol{c}_{z_i^g}^g, \quad z_i^g = \arg\min_j ||\ell_2(\mathbf{z}_{c[g]}(\boldsymbol{e}_i^f)) - \ell_2(\boldsymbol{c}_j^g)||_2,
\end{aligned}
\tag{2}
$$

where $\ell_2$ represents $\ell_2$ normalization and $\mathbf{z}_{q[g]}(\boldsymbol{e}_i^f)$ is the quantized vector from $g$-th sub-quantizer. As shown in Figure 2 (b), $\mathbf{z}_{q[g]}(\boldsymbol{e}_i^f)$ from $G$ sub-quantizers are concatenated to the full code $\mathbf{z}_q(\boldsymbol{e}_i^f) = \left[ \mathbf{z}_{q[1]}(\boldsymbol{e}_i^f), ..., \mathbf{z}_{q[G]}(\boldsymbol{e}_i^f) \right]$. Then, the code $\mathbf{z}_q(\boldsymbol{e}_i^f)$ is linearly mapped to the quantized embedding $\boldsymbol{z}_i \in \mathbb{R}^d$, which is equivalent to summing $\boldsymbol{z}_i^{q[g]} \in \mathbb{R}^d$ from $G$ sub-quantizers, i.e., $\boldsymbol{z}_i = \sum_{g=1}^G \boldsymbol{z}_i^{q[g]}$.

Given the quantized embeddings $\mathcal{Z} = \{\boldsymbol{z}_i | i = 1, ..., N_f\}$, the Neural Decoder converts them back into neural signals $\tilde{\mathcal{X}}_p^f = \{\tilde{\boldsymbol{x}}_i^f | i = 1, ..., N_f\}$. The mean squared error (MSE) loss is utilized to guide the regression. Besides, we introduce the partial-correlation constraint (Hazarika et al., 2020; Li et al., 2025a) to encourage codex-wise independence. The total loss $\mathcal{L}_f^{\mathcal{VQ}}$ for the VQ-VAE stage is:

$$
\mathcal{L}_f^{\mathcal{VQ}} = \sum_{i=1}^{N_f} [\mathcal{L}_{rgs} + \mathcal{L}_{vq} + \mathcal{L}_{pc}], \quad \mathcal{L}_{rgs} = ||\tilde{\boldsymbol{x}}_i^f - \boldsymbol{x}_i^f||_2^2, \quad \mathcal{L}_{pc} = \sum_{j=1}^{G-1} \left[ \sum_{k=j+1}^{G} \boldsymbol{z}_i^{q[j]} \cdot \boldsymbol{z}_i^{q[k]} \right],
$$

$$
\mathcal{L}_{vq} = \sum_{g=1}^{G} \left[ ||\mathbf{sg}[\mathbf{z}_{c[g]}(\boldsymbol{e}_i^f)] - \mathbf{z}_{q[g]}(\boldsymbol{e}_i^f)||_2^2 + \beta ||\mathbf{z}_{c[g]}(\boldsymbol{e}_i^f) - \mathbf{sg}[\mathbf{z}_{q[g]}(\boldsymbol{e}_i^f)]||_2^2 \right],
\tag{3}
$$

where $\mathbf{sg}$ represents the stop-gradient operation, which is an identity at the forward pass and has zero gradients. To stabilize the codex update, we use the exponential moving average strategy (Van Den Oord et al., 2017).

**DPQ-guided Mask Modeling.** BrainStratify-Fine uses DPQ-guided mask modeling to learn contextual representations. Given a sample $\mathcal{X}$, the patch tokenizer transforms it into patch embeddings $\mathcal{E}_p^f$. Around 50% of embeddings are patch-wise chosen and masked. The masked position is termed as $\mathcal{M}$. Then, a shared learnable embedding $\boldsymbol{e}_{[M]} \in \mathbb{R}^d$ is used to replace the original patch embeddings:

$$\mathcal{E}_m^f = \{\boldsymbol{e}_i^m | i = 1, ..., N_f\}, \quad \boldsymbol{e}_i^m = m_i \odot \boldsymbol{e}_{[M]} + (1 - m_i) \odot \boldsymbol{e}_i^p, \tag{4}$$

where $\delta(\cdot)$ is the indicator function and $m_i = \delta(i \in \mathcal{M})$. After that, the masked embeddings $\mathcal{E}_m^f$ will be fed into the Temporal Transformer. The output embeddings $\mathcal{E}^f$ will be used to predict the indices of the corresponding codes from the codex in the DPQ through a linear classifier:

$$p(z_i^g | \boldsymbol{e}_i^f) = \text{softmax}(\text{Linear}(\boldsymbol{e}_i^f)), \tag{5}$$

The total loss $\mathcal{L}_f^{\mathcal{M}}$ for training the MAE model is:

$$\mathcal{L}_f^{\mathcal{M}} = - \sum_{i \in \mathcal{M}} \left[ m_i \odot \sum_{g=1}^{G} \log p(z_i^g | \boldsymbol{e}_i^f) \right]. \tag{6}$$

## 4 EXPERIMENTS

### 4.1 DATASET

To validate the effectiveness of BrainStratify, we evaluate six datasets (Table 1). In addition to two public sEEG datasets (Zheng et al., 2025; Wang et al., 2024a), we further collect well-annotated Chinese word-reading & motor-intention (epidural) ECoG datasets (Appendix A). The ECoG electrodes are positioned epidurally (Liu et al., 2024) – outside the brain's dura mater rather than directly on the cortex – minimizing tissue damage compared to traditional intracranial placements (Moses et al., 2021). Besides, additional modalities (Willett et al., 2023; Sivakumar et al., 2024) are included.

Table 1: Overview of datasets used in this work.

| Name | Type | Task | # of Subjects | # of Channels | # of Trials | Trial Length | # of Classes | Total Recordings |
|------|------|------|---------------|---------------|-------------|--------------|--------------|------------------|
| Ours | ECoG | Read | 1 | 128 | $\sim$12k | 2.4s | 62 | 22 hours |
| | ECoG | Motor | 1 | 128 | $\sim$1k6 | 5s | 4 | 15 hours |
| Du-IN | sEEG | Read | 12 | 109.75 | $\sim$3k | 3s | 61 | 15 hours |
| Brain Treebank | sEEG | Listen | 10 | 124.9 | $\sim$2k | 4s | 2 | 5.5 hours |
| NPTL | MEA | Read | 1 | 128 | $\sim$10k | $\sim$10s | 39 | $\sim$20 hours |
| emg2qwerty | sEMG | Type | 8 | 32 | $\sim$4k | 4s | 98 | $\sim$3 hours |

### 4.2 IMPLEMENTATION DETAILS

**Preprocess.** We filter the sEEG/ECoG signals between 0.5Hz and 200Hz to remove low-frequency noise. Then, a notch filter of 50Hz (or 60Hz) is applied to avoid power-line interference. Next, neural signals are resampled to 400Hz and re-referenced (Li et al., 2018) according to the original setting. Finally, z-score normalization is performed on each channel to standardize data scales.

**Model Configurations.** In the Coarse-DL stage, the raw patches are first transformed into patch embeddings with $d = 256$. The following Temporal & Spatial Transformer both contain a 4-layer Transformer encoder with model dimension $d = 256$, inner dimension (FFN) $d_{ff} = 1024$, and 8 attention heads. In the Fine-DL stage, the raw patches are transformed into patch embeddings with $d = 256$, followed by an 8-layer Transformer encoder with model dimension $d = 256$, inner dimension (FFN) $d_{ff} = 1024$, and 8 attention heads. See Appendix D for more details.

**Pre-training.** All models in Coarse-DL and Fine-DL stages are trained using all recordings from each subject, excluding those reserved for validation and testing in downstream tasks. For each subject, models are trained on 1 GPU (NVIDIA Tesla V100 32GB using Python 3.11.7 and PyTorch 2.1.2 + CUDA 12.3) with data augmentation (Appendix E) for $\sim$6 hours in total.

**Fine-tuning.**  We split the task recordings into training, validation, and testing splits with a size roughly proportional to 80%, 10%, and 10%. All experiments are conducted on the same machine with the same set of random seeds. The train/validation/test splits are the same across different models. For each subject, models are trained with data augmentation (Appendix E) for ∼20 minutes. The best models are trained on the training set, selected from the validation set according to accuracy, and finally evaluated on the test set. For model comparison, we report the average and standard error values (of all subjects) on six random seeds to obtain comparable results. For subject-wise evaluation, we report the average and standard deviation values (of each subject) in Appendix O.

## 4.3 RESULTS ON CHANNEL CLUSTER

Given the sparse distribution of sEEG electrodes across the brain, we begin by identifying functional groups at a coarse level. These groups act as fundamental computational modules (Buzsáki, 2006; Silva et al., 2024), wherein channels provide complementary information to encode specific functions (e.g., vocal production) collectively. After pre-training, we compare the channel connectivity from DUET, PopT, and our method in Figure 3 (b). To standardize comparisons, we normalize connectivity values to a [0,1] range for each channel, accommodating method-specific scaling differences. The identified functional groups by our method is visualized in Figure 3 (c). And we demonstrate weighted anatomical counts (Appendix I) wrt. Desikan-Killiany atlas (Desikan et al., 2006) in Figure 3 (d).

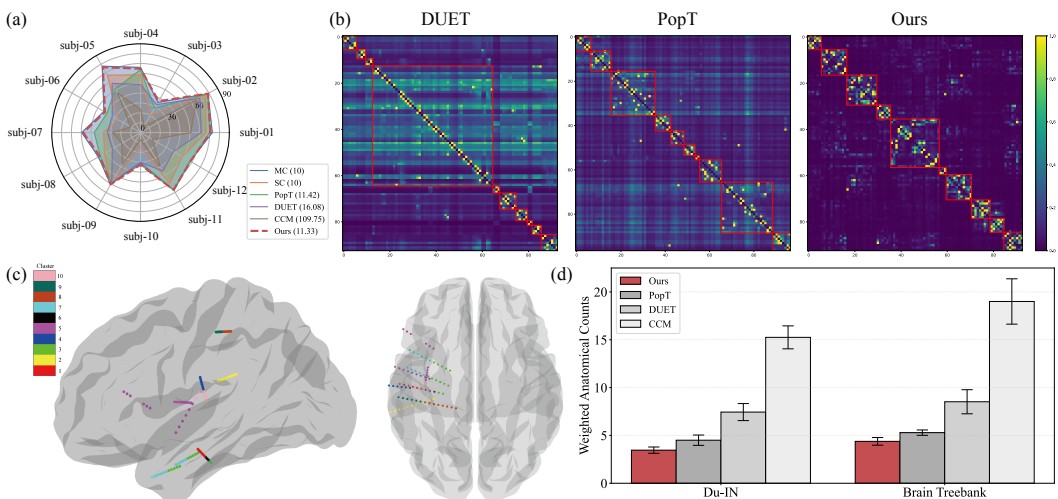

Figure 3: **Results for Coarse Disentanglement Learning. (a).** The performance on Du-IN word classification using channels selected via different strategies, with the average number of channels selected per strategy indicated in parentheses. **(b).** The channel connectivity from different methods on sEEG datasets. **(c).** The visualization of functional groups identified by our method on sEEG datasets. **(d).** The weighted anatomical counts of different methods on sEEG datasets.

Since DUET relies heavily on inter-channel correlations, we pre-train it using raw sEEG signals (without re-referencing), while using re-referenced signals instead even degrades clustering performance. As shown in Figure 3 (b), DUET struggles to reliably identify functional groups, suggesting that correlation-based metrics may not adequately capture the underlying inter-channel relationships. While PopT employs a spatial context task during pre-training, it faces convergence challenges with limited neural data. Furthermore, when pre-trained across multiple subjects, PopT underperforms our method in identifying functional groups (Figure 3 (a)), primarily due to inherent variability in neural computation across individuals. In contrast, our method effectively captures the sparse inter-channel relationships, aligning with established neuroscientific principles of functional specificity in neural processing (Buzsáki, 2006; Silva et al., 2024). Based on these findings, we employ hard clustering ($k = 10$) on the estimated channel connectivity from different methods, as illustrated in Figure 3 (b).

After clustering channels, we select and combine these groups based on their performance in downstream decoding tasks, while different backbones show similar results. We use BrainStratify-Fine (w/o pre-training) as evaluation backbone (Figure 3 (a)). Notably, while supervised strategies (i.e., SC

& MC) require 3k samples, channel selection via self-supervised strategies only requires 600 samples, highlighting the label efficiency of our method. CCM relies on static cluster embeddings, which ignore the dynamic nature of sEEG signals, resulting in unreliable functional group identification. The SC method, which fails to capture complementary information among channels, underperforms the MC strategy. In contrast, our method matches the MC baseline, validating its effectiveness.

## 4.4 RESULTS ON NEURAL DECODING

Table 2 & 3 compare our method against advanced baselines (Appendix C & D). More results on other datasets are provided in Appendix J. Our method surpasses all baselines (Table 2), highlighting DPQ's effectiveness in guiding mask modeling. H2DiLR (Wu et al., 2024) lags behind Du-IN due to the lack of mask modeling. PopT aggregates channels with [CLS] token, performing between recent foundation models (Jiang et al., 2024; Wang et al., 2024c) and EEG-CFMR (Song et al., 2022). Due to relatively stable interaction among neuronal populations, temporal models can better handle the variability of neural patterns along the temporal axis by aggregating channels into tokens before modeling temporal relationships, outperforming temporal-spatial models. Notably, this gap narrows for (epidural) ECoG data, likely stemming from inherent signal property differences. Besides, our method achieves greater improvements over Du-IN on (epidural) ECoG data compared to sEEG data, likely due to ECoG's lack of spatial resolution enhancement techniques (e.g., bi-polar re-reference).

Table 2: Results on word-reading sEEG datasets (sEEG & (epidural) ECoG). Paired T-tests are evaluated between our method and other decoding baselines. We report top-1 accuracy (%) and $1 - \texttt{SER}$ (%) across 6 random seeds for word classification and syllable sequential classification tasks, respectively; please see Appendix B for more details.

| Methods | Chan. Select. | Du-IN | | Ours | |
| --- | --- | --- | --- | --- | --- |
| | | **Word** | **Syllable** | **Word** | **Syllable** |
| LaBraM | MC | 11.78±2.70 | - | 28.33±0.98 | - |
| CBraMod | MC | 11.52±2.48 | - | 26.88±1.63 | - |
| PopT | MC | 22.55±3.26 | - | 29.30±0.63 | - |
| EEG-CFMR | MC | 45.82±4.66 | 62.63±3.77 | 42.16±2.46 | 59.59±0.42 |
| Du-IN | MC | 62.70±4.69 | 70.66±3.74 | 52.63±1.68 | 66.75±0.72 |
| H2DiLR | MC | 25.84±3.12 | 43.29±1.84 | 32.21±1.33 | 46.63±0.48 |
| BrainStra.-Fine | MC | 66.35±3.86 | **75.54±3.19** | 58.50±1.51 | **70.66±0.76** |
| BrainStra.-Fine | BrainStra.-Coarse | **66.44±3.65** | 75.36±3.17 | - | - |

† $p < 0.001$ (purple); $p < 0.01$ (pink); $p < 0.05$ (yellow); $p > 0.05$ (gray)

In Table 3, our method identifies functional groups that capture complementary neural information related to speech perception, enhancing downstream decoding. When evaluated on channels selected by our method, most models surpass the PopT baseline with SC selection. Besides, our method surpasses all baselines, further demonstrating the superiority of temporal modeling approaches.

## 4.5 RESULTS ON DISENTANGLEMENT LEARNING

To evaluate the disentanglement of the learned codexes in BrainStratify-Fine, we adopted the experiment design outlined by Metzger et al. (2023) and collected a well-annotated articulation control dataset (Figure 4 (a)). Specifically, the subject attempted (non-vocalized) movements of the jaw, lips, and tongue, resulting in ~2k trials across 6 articulation control movements (i.e., jaw open & close, lips forward & backward, and tongue up & down). Leveraging this benchmark dataset, we investigate the disentanglement of the learned codexes in relation to the basic articulation control movements.

Following the evaluation protocol commonly adopted in the representation learning based neural disentanglement field (Liu et al., 2021; Huang et al., 2024; Wang et al., 2024e), we leverage the reconstructed signals from each codex individually to train binary classification models, and report the top-1 accuracy in Figure 4 (b). Similar to the results via self-supervised disentanglement representation learning methods on neural spike recordings (Liu et al., 2021; Huang et al., 2024), the contribution of different codex to articulator control movements varies. Specifically, codex-8

Table 3: Results on Brain Treebank sEEG dataset. Paired T-tests are evaluated between our method and other decoding baselines. We report ROC-AUC across 6 random seeds for binary state classification tasks; please see Appendix B for more details.

| Methods | Chan. Select. | ROC-AUC $\pm$ ste | | | |
|---|---|---|---|---|---|
| | | Pitch | Volumn | Sent. Onset | Word Onset |
| PopT | SC | 0.74±0.03 | 0.87±0.03 | 0.90±0.01 | 0.93±0.02 |
| LaBraM | BrainStra.-Coarse | 0.72±0.02 | 0.87±0.02 | 0.90±0.02 | 0.94±0.01 |
| CBraMod | BrainStra.-Coarse | 0.70±0.02 | 0.84±0.03 | 0.90±0.02 | 0.94±0.01 |
| PopT | BrainStra.-Coarse | 0.74±0.02 | 0.88±0.02 | 0.94±0.01 | 0.96±0.01 |
| EEG-CFMR | BrainStra.-Coarse | 0.77±0.02 | 0.89±0.02 | 0.92±0.02 | 0.94±0.01 |
| Du-IN | BrainStra.-Coarse | 0.78±0.02 | 0.90±0.02 | 0.94±0.01 | 0.97±0.01 |
| H2DiLR | BrainStra.-Coarse | 0.77±0.02 | 0.88±0.02 | 0.88±0.02 | 0.90±0.01 |
| BrainStra.-Fine | BrainStra.-Coarse | **0.79±0.02** | **0.91±0.02** | **0.95±0.02** | **0.98±0.01** |

† $p < 0.001$ (purple); $p < 0.01$ (pink); $p < 0.05$ (yellow); $p > 0.05$ (gray)

consistently encodes articulation control information across articulation control tasks, where codex-7, codex-2&3, and codex-1&4&5 are primarily involved in jaw, lips, and tongue control, respectively.

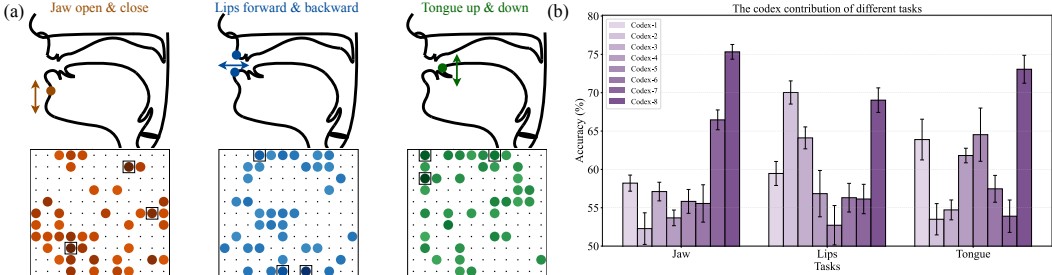

Figure 4: **Results for Fine Disentanglement Learning. (a).** The channel tuning map across articulators during attempted movements. **(b).** The codex contribution analysis across articulator control tasks (i.e., jaw & lips & tongue). Since the spatial resolution of ECoG data seems insufficient to discriminate neural encoding between the front and back tongue at explored by Metzger et al. (2023), our analysis focuses primarily on the codex's encoding of the broader articulators: the jaw, tongue, and lips. The reconstruction loss caused by keeping any codex to reconstruct the original signal is close to 1, with minor differences.

## 5 LIMITATIONS

Given the sparse distribution of sEEG electrodes across the brain, BrainStratify is currently limited to identifying functional modules rather than full functional networks. Nevertheless, these modules serve as fundamental building blocks for such networks. Extending our framework to infer dynamic network topology would enrich its neuroscientific insight if denser sEEG electrode coverages become available and models are refined to capture time-lagged spatial relationships.

Due to the scarcity of public intracranial neural datasets, BrainStratify is currently evaluated on six datasets (e.g., vocal production, speech perception, etc.). Despite this limitation, the framework's design affords clear interpretability grounded in brain organization, ensuring a solid neuroscience foundation (Buzsáki, 2006; Silva et al., 2024; Metzger et al., 2023). We anticipate our method to generalize to other cognitive states (e.g., image perception) as more data becomes publicly available.

## 6 DISCUSSION

In the field of neural encoding, neuroscience researchers design cognitive experiments and collect intracranial datasets to study the organization of cognitive functions (Bouchard et al., 2013; Mesgarani et al., 2014). Specifically, they aim to break down cognitive function and identify the stable and precise information encoded by each brain region (e.g., vSMC for vocal production, STG for speech perception, etc.). However, large-scale labeling in clinical settings is often prohibitively costly or unfeasible (Angrick et al., 2021; Singh et al., 2025), which contradicts the large amount of labeled data required by supervised channel-selection methods (e.g., SC and MC). And this issue may prevent researchers from fully exploiting their valuable and hard-won datasets, thus delaying the discovery process in cognitive state decoding. For example, in some early speech decoding studies (Angrick et al., 2021), researchers were not aware of the importance of channel selection based on functional similarity, resulting in not fully exploiting the potential of the collected data.

In this work, we present BrainStratify, a practical toolbox for intracranial neural decoding in real-world clinical settings. BrainStratify leverages abundant unlabeled data to address two fundamental problems in intracranial modeling: (1) data-efficient channel selection in intracranial sEEG recordings; (2) representation learning via self-supervision to boost performance across diverse decoding paradigms (e.g., classification, sequential classification, regression). By enabling neuroscience researchers to fully exploit their valuable and hard-won datasets, our framework will potentially accelerate the discovery process in cognitive state decoding.

The core design of BrainStratify-Coarse is largely consistent with studies on the neural encoding of cognitive function organization (Bouchard et al., 2013; Mesgarani et al., 2014), which aims to break down cognitive function and identify the stable and precise information encoded by each brain region (e.g., vSMC for vocal production, STG for speech perception, etc.). For such cognitive-decoding tasks, our method aims to estimate the precise functional boundaries of these modules, as demonstrated in Figure 3, thus avoiding the involvement of irrelevant channels to reduce performance. However, decoding tasks are not limited to cognitive functions; there are other types of decoding tasks, such as epilepsy detection (Zhang et al., 2023; Jiang et al., 2024) and sleep staging (Liu & Jia, 2023; Vallat & Walker, 2021). Therefore, our model is not capable of handling all types of decoding tasks, but focuses on decoding tasks related to cognitive functions (e.g., speech decoding).

BrainStratify-Fine aims to understand the fine-grained neural components within target functional groups via disentanglement representation learning. Due to the lack of more public datasets suitable for evaluating disentanglement, we presently confine our claim to the speech decoding field. That said, we believe that our model can be extended to other similar decoding tasks. We preliminarily evaluated our framework on an ECoG motor-intention task and an sEMG typing task. The positive decoding (not disentanglement) results on these non-language tasks suggest promising potential for broader extrapolation, and non-language tasks are worth systematically exploring if the corresponding datasets for studying encoding mechanisms are available.

## 7 CONCLUSION

This paper proposes BrainStratify, a novel speech decoding framework for intracranial recordings, enhanced by Coarse-to-Fine neural disentanglement. Comprehensive experiments demonstrate that BrainStratify-Coarse reliably identifies functional groups from sEEG signals, surpassing existing channel clustering baselines. In addition, BrainStratify-Fine effectively decouples neural components within target functional groups, achieving superior performance across intracranial neural modalities (e.g., sEEG, (epidural) ECoG) compared to advanced neural decoding baselines. Overall, our approach – inspired by neuroscience findings – is suitable for decoding speech from intracranial neural signals, advancing toward clinically viable and transparent neuroprosthetic systems.

THE USAGE OF LLMs

Our writing process was assisted by DeepSeek-R1 (Guo et al., 2025), which was used to polish textual clarity. Brief paragraphs were provided to the model, and its output was critically evaluated before relevant revisions were adopted for the final version.

REPRODUCIBILITY STATEMENT

Code to train models and reproduce the results is submitted as part of the supplementary materials and can be accessed here: https://anonymous.4open.science/r/BrainStratify-07E6.

ETHICS STATEMENT

Experiments that contribute to this work were approved by IRB. All subjects consent to participate. All electrode locations are exclusively dictated by clinical considerations.

Our informed consent signing process is as follows:

1. If the experimental participants are adults and have full civil capacity, we will ask them to sign a written informed consent after the participants have fully informed consent;

2. If the experimental participants are minors or do not have full civil capacity, we will ask the participant's legal guardian to sign a written informed consent after the participants and their legal guardians have fully informed consent.

Our informed consent form includes the following points:

1. Contact information of research institutions and researchers;

2. Research direction and purpose;

3. Risks involved in the research;

4. Personal information, data and usage methods to be used in the research;

5. Privacy protection statement (all personal identification information (PII) will not be disclosed);

6. Data storage statement (retained after deleting all personal identification information (PII));

7. Voluntary statement of participants;

8. Statement that participants can withdraw unconditionally at any time.

Our data storage and protection procedures include the following processes:

1. Our data collection, transfer, and analysis tasks are only completed by researchers who have signed relevant confidentiality agreements;

2. The collected raw data will be copied twice as soon as possible, one copy to a storage computer that is not connected to the Internet and encrypted, and the other copy to a mobile hard disk and encrypted and stored offline;

3. The use of the data is only authorized to the research leader and the main researchers (less than 5 people), among which the main researchers can only access data that does not contain personal identification information (PII);

4. After the study is completed, all personal identification information (PII) on both nodes (storage computer, mobile hard disk) will be deleted immediately.

To prevent unauthorized access or possible data leakage, we use double encryption on the storage computer, that is, a static password and a dynamic password (received by mobile phone or email); physical isolation is used on the mobile hard disk, that is, it is locked in a filing cabinet, and the key is only kept by the research leader and the main researchers.

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

## A  EXPERIMENT DESIGN

### A.1  WORD-READING EPIDURAL ECoG DATASET

Due to the lack of open-source intracranial neural datasets related to speech, we follow the experimental design outlined in Du-IN (Zheng et al., 2025) to collect a well-annotated Chinese word-reading (epidural) ECoG dataset, including one subject (female; aged 67) who has lost her ability to communicate or perform daily tasks due to amyotrophic lateral sclerosis (ALS).

We developed a minimally invasive BCI (Branco et al., 2023; Liu et al., 2024) with an $11 \times 12$ (epidural) ECoG grid (50.30mm × 52.21mm) above the ventral sensorimotor cortex (vSMC) to restore the speech functions of that subject, as shown in Figure 5 (a). With wireless powering and neural data transmission, this system enables real-time speech neuroprosthesis in home use. After excluding the four corner electrodes, we analyzed neural recordings from the remaining 128 channels, as shown in Figure 5 (b). All electrodes (except No.23 and No.25) exhibit consistently low impedance levels during neural recordings, ensuring robust signal fidelity.

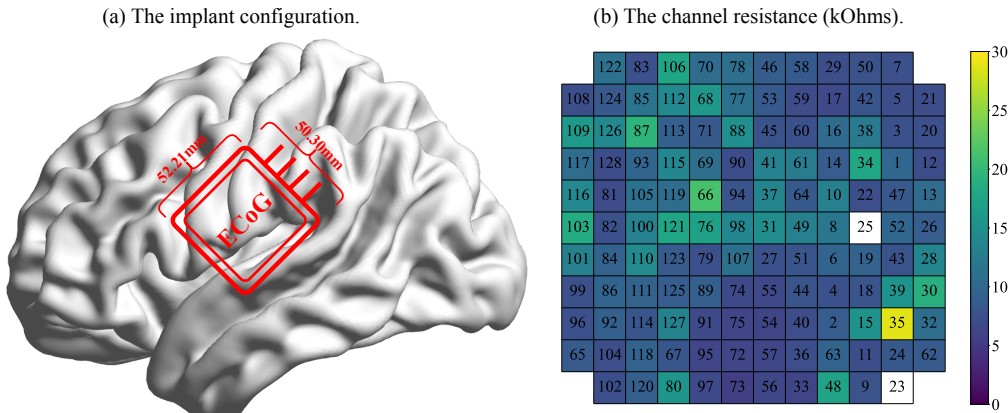

(a) The implant configuration.          (b) The channel resistance (kOhms).

Figure 5: **Overview of ECoG configuration. (a).** The implant configuration. Our developed (epidural) ECoG is placed above vSMC, which is involved in vocal production (Silva et al., 2024). **(b).** The channel resistance. Electrodes at the four corners are excluded for downstream analysis.

In the word-reading task, the subject attempts to speak individual words from the pre-defined word-set (including 62 words as detailed in Table 4) while we record her brain activity (measured by ECoG).

All data are collected as a series of "blocks" (42 blocks in total), with each block lasting about 20 minutes and consisting of multiple trials. During each block of this task, all words (from the pre-defined word-set) are presented individually 5 times, leading to a total of 310 trials.

Each trial in a block of this task starts with one word shown on the screen in white text. After 0.2 seconds, the text will turn green and remain on the screen for 2.2 seconds. This color transition from white to green represents the go cue for each trial, and the subject is instructed to speak the word aloud as soon as the text turns green. Afterward, the text will be replaced with a blank screen with a centered cross. After 0.8 seconds, the task continues to the next trial. The word presentation order is randomized within each task block.

Besides, we also collected non-task recordings of subjects in their daily life. There are roughly 8 hours of non-task recordings during wakefulness. In summary, for each subject, we collect about 22 hours of sEEG recordings, of which 14 hours are task recordings.

### A.2  MOTOR-INTENTION EPIDURAL ECoG DATASET

We collected neural data from one paralyzed subject (male; aged 30) who has lost his motor function in all four limbs due to spinal cord injuries (SCI). The subject underwent a surgical procedure to implant the (epidural) ECoG grid, above the subregion of the left precentral gyrus responsible for

motor control (i.e., hand, leg), in his brain. All channels exhibit consistently low impedance levels during neural recordings, ensuring robust signal fidelity.

In the motor-intention task, the subject attempted to move his limbs from the pre-defined action-set while we recorded his brain activity (measured by ECoG). Specifically, word cues are replaced with the actions from the pre-defined action-set: (1) left-finger: rotate the left index finger; (2) right-hand: clench the right fist; (3) right-elbow: bend the right elbow; (4) right-leg: bend the right knee. And the duration of the green prompt is extended to 4 seconds from 2 seconds. Besides, we also collected non-task recordings of the subject in his daily life. There are roughly 10 hours of non-task recordings during wakefulness. In summary, for the subject, we collected about 15 hours of (epidural) ECoG recordings, of which 5 hours are task recordings.

Table 4: The Chinese words and English translations in the word-reading (epidural) ECoG dataset.

| Words | Translations | Words | Translations | Words | Translations |
|---|---|---|---|---|---|
| 的 | of | 对 | right | 在 | exist |
| 把 | handle | 是 | be | 要 | want |
| 和 | and | 你 | you | 这 | this |
| 去 | go | 有 | have | 没 | without |
| 他 | he | 看 | look | 我 | I |
| 给 | give | 不 | no | 都 | all |
| 就 | at once | 帮 | help | 好 | good |
| 找 | find | 陪 | accompany | 热 | hot |
| 冷 | cold | 人 | people | 想 | think |
| 吗 | ? | 出 | out | 医生 | doctor |
| 可以 | can | 睡觉 | sleep | 说话 | speak |
| 休息 | rest | 问题 | question | 家人 | family |
| 谢谢 | thanks | 朋友 | friend | 吃饭 | eat |
| 手机 | cell phone | 喝水 | drink water | 心情 | mood |
| 厕所 | toilet | 快乐 | happy | 困难 | difficulty |
| 紧急 | urgent | 护士 | nurse | 感觉 | feel |
| 舒服 | comfortable | 电脑 | computer | 坐着 | sitting |
| 躺下 | lie down | 洗澡 | bath | 现在 | now |
| 书籍 | book | 医院 | hospital | 衣服 | clothes |
| 一起 | together | 散步 | walk | 呼吸 | breathe |
| 非常 | very | 回家 | go home | - | - |

## B TASK DETAILS

In experiments, we evaluate multiple neural decoding tasks across diverse datasets: two public sEEG datasets (Zheng et al., 2025; Wang et al., 2024a), our collected word-reading & motor-intention (epidural) ECoG datasets, and two public datasets from other modalities (i.e., Micro-Electrode Array (MEA), surface ElectroMyoGraphy (sEMG)).

### B.1 DU-IN SEEG DATASET

We follow the same task specification in Du-IN (Zheng et al., 2025). Each trial lasts 3 seconds.

**Word Reading.** The subject speaks aloud individual words from the pre-defined word-set (including 61 words) while his neural activity and voice are simultaneously recorded. Labels are balanced in the experiment design.

We follow the evaluation protocol in Du-IN (Zheng et al., 2025). Since this task is a classification (CLS) task, we flatten embeddings and add a linear head after either pre-trained or randomly initialized models. Training employs cross-entropy loss, with results quantified using top-1 accuracy scores.

Besides, following Willett et al. (2023), we evaluate the 49-syllable sequential classification (CTC) task. Considering the difference between English and Chinese, we utilize syllables from Pinyin (Wang, 1973), a widely adopted phonetic representation system based on the Latin alphabet, as basic units that have the potential to support open-set speech decoding tasks. The set of 49 syllables includes:

- 1 CTC blank token (i.e., "-"),
- 1 silence token (i.e., "|"),
- 23 initial syllable tokens,
- 24 final syllable tokens.

Take the pre-determined word "电脑" (i.e., "computer") for example, the corresponding syllable label sequence $y$ is ["|", "d", "i", "an", "|", "n", "ao", "|"]. Since this task is a sequential classification (CTC) task, we add a linear head after either pre-trained or randomly initialized models. Training employs connectionist temporal classification (CTC) loss, with results quantified using syllable error rate scores. Specifically, the syllable error rate (SER) is computed based on the syllable edit distances, which is widely adopted in neural decoding research (Fan et al., 2023; Willett et al., 2023; Metzger et al., 2023). To align with the trend of top-1 accuracy, we present results as (1-SER) rather than raw SER values.

### B.2 BRAIN TREEBANK SEEG DATASET

While adopting the same task specification in PopT (Chau et al., 2024), we narrowed the analysis window to [-2.0,2.0]s around word onset (vs. PopT's [-2.5,2.5]s), yielding 4-second neural activity per trial.

**Pitch.** The pitch of a given word is extracted using Librosa's `piptrack` function over a Mel-spectrogram (sampling rate 48,000 Hz, FFT window length of 2048, hop length of 512, and 128 mel filters). For this task, for a given session, the positive examples consist of words in the top quartile of pitch, and the negative examples are the words in the bottom quartile.

**Volume.** The volume of a given word is computed as the average intensity of root-mean-square (RMS) (`rms` function, frame and hop lengths 2048 and 512 respectively). For this task, for a given session, the positive examples are the words in the top quartile of volume, and the negative examples are the words in the bottom quartile.

**Sent. Onset (Sentence Onset).** The negative examples are intervals of activity from 1s periods during which no speech occurs in the movie. The positive examples are intervals of brain activity that correspond with hearing the first word of a sentence.

**Word Onset (Speech vs. Non-speech).** The negative examples are intervals of activity from 1s periods during which no speech occurs in the movie. The positive examples are intervals of brain activity that correspond with dialogue being spoken in the stimuli movie.

For each task, we follow the evaluation protocol in PopT (Chau et al., 2024), using the specified movie for downstream classification. Since these tasks are binary classification (CLS) tasks, we flatten embeddings and add a linear head after either pre-trained or randomly initialized models. Training employs binary cross-entropy (BCE) loss, with results quantified using ROC-AUC scores.

### B.3 WORD-READING (EPIDURAL) ECoG DATASET

The experiment design is discussed in Appendix A. Each trial lasts 2.4 seconds.

**Word Reading.** The subject attempts to speak individual words from the pre-defined word-set while her neural activity is recorded. Labels are balanced in the experiment design.

We follow the evaluation protocol in Du-IN (Zheng et al., 2025). Since this task is a classification (CLS) task, we flatten embeddings and add a linear head after either pre-trained or randomly initialized models. Training employs cross-entropy loss, with results quantified using top-1 accuracy scores.

Besides, following the aforementioned evaluation protocol for the Du-IN sEEG dataset, we evaluate the 49-syllable sequential classification (CTC) task. Training employs connectionist temporal classification (CTC) loss, with results quantified using syllable error rate scores.

### B.4 MOTOR-INTENTION (EPIDURAL) ECoG DATASET

The experiment design is discussed in Appendix A. Each trial lasts 5 seconds.

**Motor Intention.** The subject attempts to move his limbs from the pre-defined action-set while is neural activity is recorded. Labels are balanced in the experiment design.

Since this task is a classification (CLS) task, we flatten embeddings and add a linear head after either pre-trained or randomly initialized models. Training employs cross-entropy loss, with results quantified using top-1 accuracy scores.

### B.5 NPTL MEA DATASET

We follow the same task specification in Willett et al. (2023). Each trial lasts ∼10 seconds.

**Sentence Reading.** The subject attempts to speak selected sentences from the pre-defined sentence-set (covering 125,000 words) without repetition while his neural activity is recorded.

We follow the evaluation protocol in Willett et al. (2023). Since this task is a sequential classification (CTC) task, we flatten embeddings according to the specified flatten window and add a linear head after either pre-trained or randomly initialized models. Training employs connectionist temporal classification (CTC) loss, with results quantified using phoneme error rate scores (PER). To align with the trend of top-1 accuracy, we present results as $(1 - \text{PER})$ rather than raw PER values.

### B.6 EMG2QWERTY SEMG DATASET

We follow the same task specification in Sivakumar et al. (2024). Each trial lasts 4 seconds.

**Character Typing.** The subjects type characters according to the selected sentences from the pre-defined sentence-set without repetition while their sEMG signals are recorded.

We follow the evaluation protocol in Sivakumar et al. (2024). Since this task is a sequential classification (CTC) task, we flatten embeddings according to the specified flatten window and add a linear head after either pre-trained or randomly initialized models. Training employs connectionist temporal classification (CTC) loss, with results quantified using character error rate scores (CER). To align with the trend of top-1 accuracy, we present results as $(1 - \text{CER})$ rather than raw CER values.

## C    BASELINE DETAILS

### C.1    CHANNEL CLUSTER BASELINES

In experiments, we compare our model to the advanced channel cluster methods (Chen et al., 2025; Qiu et al., 2024) in time series analysis. The details of these baseline models are given here:

- **CCM** (Chen et al., 2025): A time-series forecasting model that learns channel clustering based on intrinsic similarities and creates prototype embeddings for each cluster via a cross-attention mechanism. Since sEEG is a unique type of time series, this model is suitable to serve as a baseline for comparison.

- **DUET** (Qiu et al., 2024): A time-series forecasting model that captures the relationships among channels in the frequency domain through metric learning and applies sparsification to mitigate the adverse effects of noisy channels. Since sEEG is a unique type of time series, this model is suitable to serve as a baseline for comparison.

The detailed implementations of these baseline models are given here:

- As the CCM method (Chen et al., 2025) relies heavily on the correlation-based similarities among channels, we use raw sEEG signals of each subject (before bi-polar re-reference (Li et al., 2018)) as inputs. Bi-polar re-reference disrupts these correlations, causing CCM to fail convergence reliably. Because CCM does not generate an intermediate inter-channel similarity matrix (i.e., $\mathcal{P}$ in this work), we directly utilize the assigned clusters for downstream channel selection.

- Like CCM, the DUET method (Qiu et al., 2024) relies on raw inter-channel correlations. We use raw sEEG signals of each subject (before bi-polar re-reference) as inputs, ensuring stable model convergence. In practice, DUET produces an intermediate inter-channel similarity matrix (i.e., $\mathcal{P}$ in this work) and integrates this matrix via a masked attention mechanism, enhancing time series forecasting. Therefore, we directly apply spectral cluster (Ng et al., 2001) on this matrix to derive channel clusters for downstream channel selection.

Although PopT (Chau et al., 2024) was originally proposed as an sEEG foundation model, we also compare our model with it. PopT introduces channel connectivity based on the pre-trained model, which provides an alternative to conventional coherence analysis. Similar to our model, we use bi-polar (or Laplacian) re-referenced sEEG signals as inputs. Since PopT struggles to converge with limited data, we pre-train PopT with sEEG signals from all available subjects, which is the original setting (Chau et al., 2024). After computing channel connectivity based on the pre-trained model, we apply spectral cluster (Ng et al., 2001) on this matrix to derive channel clusters for downstream channel selection.

### C.2    NEURAL DECODING BASELINES

In experiments, we compare our model to the existing supervised or self-supervised neural decoding methods (Jiang et al., 2024; Wang et al., 2024c; Chau et al., 2024; Song et al., 2022; Zheng et al., 2025; Wu et al., 2024) on brain signals. The details of these baseline models are given here:

- **LaBraM** (Jiang et al., 2024): A self-supervised model for EEG recordings that learns generic representations with tremendous EEG data. LaBraM models temporal and spatial dependencies simultaneously, serving as an EEG foundation model. Since the spatial embeddings are originally pre-defined according to the EEG caps, we replace the learnable spatial embeddings with hard-coded spatial embeddings from PopT (Chau et al., 2024) to enable multi-subject pre-training under the sEEG setting. Since the data modes of EEG and sEEG are similar, this model is suitable to serve as a baseline for comparison.

- **CBraMod** (Wang et al., 2024c): A self-supervised model for EEG recordings that captures the heterogeneity between temporal and spatial dependencies. CBraMod combines a criss-cross attention mechanism with asymmetric conditional positional encoding (ACPE) to effectively model temporal-spatial dependencies among EEG patches. CBraMod serves as an EEG foundation model, achieving SOTA performance on various EEG tasks. Since the

data modes of EEG and sEEG are similar, this model is suitable to serve as a baseline for comparison.

- **PopT** (Chau et al., 2024): A self-supervised model for sEEG that learns population-level codes for arbitrary ensembles of neural recordings at scale. PopT stacks on top of pre-trained temporal embeddings and enhances downstream decoding by enabling the learned aggregation of multiple spatially sparse channels. PopT serves as an sEEG foundation model, achieving SOTA performance on Brain Treebank (Wang et al., 2024a). As a foundation model in the sEEG pre-training field, this model is suitable to serve as a baseline for comparison.

- **EEG-CFMR** (Song et al., 2022): A supervised model for EEG that consists of both CNN module and Transformer module, to encapsulate local and global features in a unified EEG classification framework. EEG-CFMR is mainly designed for EEG-based motor imagination tasks. Since the data modes of EEG and sEEG are similar, this model is suitable to serve as a baseline for comparison.

- **Du-IN** (Zheng et al., 2025): A self-supervised model for sEEG-based speech decoding that learns contextual embeddings based on region-level tokens through discrete codex-guided mask modeling. Du-IN achieves SOTA performance on sEEG-based speech decoding using the Du-IN dataset (Zheng et al., 2025). As a strong baseline in sEEG-based speech decoding, this model is suitable to serve as a baseline for comparison.

- **H2DiLR** (Wu et al., 2024): A self-supervised model for sEEG-based tone decoding that disentangles and learns both the homogeneity and heterogeneity from intracranial sEEG recordings across multiple subjects. H2DiLR achieves SOTA performance on sEEG-based tone decoding using the sEEG dataset from Feng et al. (2023). As a strong baseline in sEEG-based tone decoding (part of speech decoding), this model is suitable to serve as a baseline for comparison.

The detailed implementations of these baseline models are given here:

- For the LaBraM method (Jiang et al., 2024), the hyperparameters are the same as the original implementation of the LaBraM-Base model. In practice, foundation models are pre-trained on massive neural datasets. Therefore, their architectures remain fixed after the pre-training stage, limiting post-hoc modifications. The data samples are resampled to the specified sampling rate (i.e., 200 Hz).

- For the CBraMod method (Wang et al., 2024c), the hyperparameters are the same as the original implementation of the CBraMod model. The data samples are resampled to the specified sampling rate (i.e., 200 Hz).

- For the PopT method (Chau et al., 2024), the hyperparameters are the same as the original implementation of the PopT model. The data samples are resampled to the specified sampling rate (i.e., 2048 Hz).

- For the EEG-CFMR method (Song et al., 2022), the hyperparameters are the same as the original implementation of the EEG-CFMR model. The data samples are resampled to the specified sampling rate (i.e., 250 Hz).

- For the Du-IN method (Zheng et al., 2025), the hyperparameters are the same as the original implementation of the Du-IN model. The data samples are resampled to the specified sampling rate (i.e., 1000 Hz).

- For the H2DiLR method (Wu et al., 2024), the hyperparameters are the same as the original implementation of the H2DiLR model. The data samples are resampled to the specified sampling rate (i.e., 1000 Hz).

When evaluating the decoding performance of these baseline models, we follow the same experiment setup of our model; see Appendix D and Appendix E for more details.

For the self-supervised methods, the pre-training setup follows the original setup of each model:

- For the LaBraM model, we include neural recordings from all available subjects within each dataset for pre-training. The data samples are 4 seconds.

- For the CBraMod model, we include neural recordings from all available subjects within each dataset for pre-training. The data samples are 4 seconds.

- For the PopT model, we include neural recordings from all available subjects within each dataset for pre-training. The data samples are 4 seconds.

- For the Du-IN model, we include neural recordings from each subject for pre-training. The data samples are 4 seconds.

- For the H2DiLR model, we include neural recordings from all available subjects within each dataset for pre-training. The data samples are 4 seconds.

# D  MODEL DETAILS

## D.1  BRAINSTRATIFY-COARSE

The BrainStratify-Coarse model (Table 5) is a general architecture for sEEG-based functional group identification, as shown in Figure 2 (a). The architecture of BrainStratify-Coarse contains three parts: (1) Temporal Convolution, (2) Temporal & Spatial Transformer, and (3) Channel Cluster Module. During the pre-training stage, one additional "Context Classification (CLS) Head" is added after the "Spatial Transformer" for spatial context classification.

**Spatial Context Task.**    Since sEEG channels capture local and depth information from different brain regions, their recordings inherently capture unique neural information with minimal overlap. This makes the spatial context task better suited for learning inter-channel relationships compared to mask-based reconstruction approaches (Jiang et al., 2024; Wang et al., 2024c). To ensure balanced label distribution, we designate only 10% of unreplaced channels as positive samples during pre-training. For all subjects used in this work, the model converges rapidly to $\geq 95\%$.

**Channel Cluster Module.**    After pre-training with the spatial context task, we calculate the channel connectivity $\mathcal{P} \in \mathbb{R}^{C \times C}$ following Algorithm 1. Then, spectral cluster (Ng et al., 2001) is applied to group channels into functional clusters, using scikit-learn's (Pedregosa et al., 2011) `cluster.SpectralClustering` with default function arguments.

Table 5: The hyperparameters for BrainStratify-Coarse training.

| Module | Name | Value |
|---|---|---|
| Temporal Convolution | # of Input Channels | {1,128,128} |
| | # of Output Channels | {128,128,128} |
| | Kernel Size | {9,9,3} |
| | Stride | {5,5,2} |
| | Padding | {4,4,1} |
| | Flatten Window | 2 |
| Temporal Transformer | # of Transformer Layers | 4 |
| | Hidden Size | 256 |
| | MLP Size | 1024 |
| | MLP Dropout Ratio | {0.2,0.} |
| | # of Attention Heads | 8 |
| | Attention Head Size | 64 |
| | Attention Dropout Ratio | 0.2 |
| Spatial Transformer | # of Transformer Layers | 4 |
| | Hidden Size | 256 |
| | MLP Size | 1024 |
| | MLP Dropout Ratio | {0.2,0.} |
| | # of Attention Heads | 8 |
| | Attention Head Size | 64 |
| | Attention Dropout Ratio | 0.2 |
| Context CLS Head | Linear Projection | $256 \to 1$ |
| Optimizer | Batch Size | 32 |
| | Maximum Learning Rate | 3e-4 |
| | Minimum Learning Rate | 5e-6 |
| | Learning Rate Scheduler | Cosine |
| | Optimizer Type | AdamW |
| | Adam $\beta$ | $(0.9, 0.99)$ |
| | Weight Decay | 0.05 |
| | Total Epochs | 100 |
| | Warm-up Epochs | 10 |

---

**Algorithm 1** The calculation of channel connectivity $\mathcal{P} \in \mathbb{R}^{C \times C}$.

---

**Require:** $\{\mathcal{X}_i \in \mathbb{R}^{C \times T} | i = 1, ..., N_{\text{samples}}\}$    $\triangleright N_{\text{samples}}$ is the number of samples.

  $\mathcal{P} \leftarrow \mathbf{0}_{C \times C}$               $\triangleright \mathcal{P} \in \mathbb{R}^{C \times C}$ is initialized as 0s.

  **while** $i \leq N_{\text{samples}}$ **do**

    $\hat{\mathcal{P}} \leftarrow \text{model}(\mathcal{X}_i)$       $\triangleright \hat{\mathcal{P}} \in \mathbb{R}^{N_{\text{layer}} \times N_{\text{head}} \times C \times C}$ is spatial attention scores.

    $\hat{\mathcal{P}} \leftarrow \text{mean}(\hat{\mathcal{P}}, \text{axes} = [0, 1])$  $\triangleright \hat{\mathcal{P}} \in \mathbb{R}^{C \times C}$ is averaged across [layer,head]-dimensions.

    $\mathcal{P} \leftarrow \mathcal{P} + \hat{\mathcal{P}}/N_{\text{samples}}$

  **end while**

---

## D.2 BRAINSTRATIFY-FINE

After identifying coarse-grained functional groups, BrainStratify-Fine aims to further identify fine-grained neural components through decoupled product quantization (DPQ). The three-stage training framework of BrainStratify-Fine is illustrated in Figure 6. "Neural Encoder" is shared across BrainStratify-Fine variants. The hyperparameters of Neural Encoder are shown in Table 6.

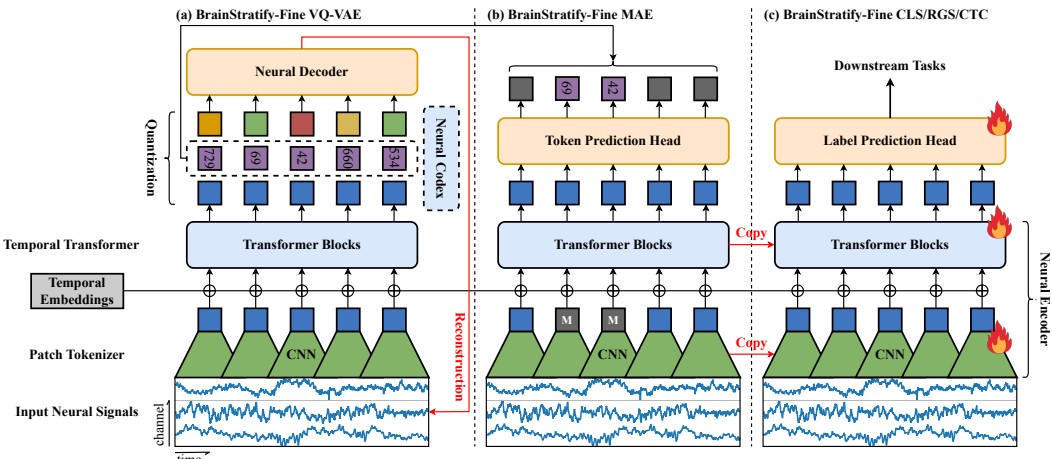

Figure 6: **An overview of the three-stage training pipeline of BrainStratify-Fine. (a).** Learning discrete neural codex in the BrainStratify-Fine VQ-VAE by reconstructing the original neural signals. **(b).** Mask modeling pre-training of Neural Encoder in the BrainStratify-Fine MAE. **(c).** Fine-tuning the pre-trained Neural Encoder with an MLP head for various downstream decoding tasks.

Table 6: The hyperparameters of Neural Encoder.

| Module | Name | Value |
|---|---|---|
| Patch Tokenizer | Linear Projection | $C \rightarrow C$ |
| | # of Input Channels | $\{C,256,256,256\}$ |
| | # of Output Channels | $\{256,256,256,256\}$ |
| | Kernel Size | $\{9,3,3,3\}$ |
| | Stride | $\{5,2,2,2\}$ |
| | Padding | $\{4,1,1,1\}$ |
| Temporal Transformer | # of Transformer Layers | 8 |
| | Hidden Size | 256 |
| | MLP Size | 1024 |
| | MLP Dropout Ratio | $\{0.2,0.\}$ |
| | # of Attention Heads | 8 |
| | Attention Head Size | 64 |
| | Attention Dropout Ratio | 0.2 |

### D.2.1 BRAINSTRATIFY-FINE VQ-VAE

The BrainStratify-Fine VQ-VAE model (Table 7) contains three parts: (1) Neural Encoder, (2) Vector Quantizer, and (3) Neural Decoder.

### D.2.2 BRAINSTRATIFY-FINE MAE

The BrainStratify-Fine MAE model (Table 8) contains two parts: (1) Neural Encoder, and (2) Token Classification (CLS) Head. The architecture of Neural Encoder is shown in Table 6. It's worth noting that when training BrainStratify-Fine MAE, the weights of "Neural Encoder" are randomly initialized, instead of being loaded from the pre-trained BrainStratify-Fine VQ-VAE model.

### D.2.3 BRAINSTRATIFY-FINE CLS

The BrainStratify-Fine CLS model (Table 9) is designed for the classification (CLS) task – decode the corresponding label $y$ from a sequence of raw neural signals $\mathcal{X}$. The architecture of BrainStratify-Fine CLS contains two parts: (1) Neural Encoder, and (2) Label Classification (CLS) Head.

### D.2.4 BRAINSTRATIFY-FINE RGS

The BrainStratify-Fine RGS model (Table 11) is designed for the regression (RGS) task – decode the corresponding label sequence $y$ from a sequence of raw neural signals $\mathcal{X}$. The architecture of BrainStratify-Fine RGS contains two parts: (1) Neural Encoder, and (2) Label Regression (RGS) Head.

### D.2.5 BRAINSTRATIFY-FINE CTC

The BrainStratify-Fine CTC model (Table 11) is designed for the sequential classification (CTC) task – decode the corresponding label sequence $y$ from a sequence of raw neural signals $\mathcal{X}$; see Appendix B for more details. The architecture of BrainStratify-Fine CTC contains two parts: (1) Neural Encoder, and (2) Label Classification (CLS) Head.

Table 7: The hyperparameters for BrainStratify-Fine VQ-VAE training.

| Module | Sub-Module | Name | Value |
|---|---|---|---|
| Neural Encoder | - | - | - |
| Vector Quantizer | - | # of Groups | 4 |
| | | Codex Size per Group | $256 \times 64$ |
| | | Embedding-to-Codex Projection | $256 \rightarrow 256(\text{Tanh}) \rightarrow 64$ |
| | | Codex-to-Embedding Projection | $64 \rightarrow 256$ |
| Neural Decoder | Temporal Transformer | # of Transformer Layers | 4 |
| | | Hidden Size | 256 |
| | | MLP Size | 1024 |
| | | MLP Dropout Ratio | $\{0.2, 0.\}$ |
| | | # of Attention Heads | 8 |
| | | Attention Head Size | 64 |
| | | Attention Dropout Ratio | 0.2 |
| | Time RGS Head | # of Input Channels | $\{256,256,256,256\}$ |
| | | # of Output Channels | $\{256,256,256,256\}$ |
| | | Kernel Size | $\{3,3,3,9\}$ |
| | | Stride | $\{2,2,2,5\}$ |
| | | Padding | - |
| | | Output Padding | - |
| | | Linear Projection | $256 \rightarrow C$ |
| Optimizer | - | Batch Size | 64 |
| | | Maximum Learning Rate | 3e-4 |
| | | Minimum Learning Rate | 5e-5 |
| | | Learning Rate Scheduler | Cosine |
| | | Optimizer Type | AdamW |
| | | Adam $\beta$ | $(0.9, 0.99)$ |
| | | Weight Decay | 0.01 |
| | | Total Epochs | 100 |
| | | Warm-up Epochs | 10 |

Table 8: The hyperparameters for BrainStratify-Fine MAE training.

| Module | Sub-Module | Name | Value |
|---|---|---|---|
| Neural Encoder | - | - | - |
| Token CLS Head | - | # of Classification Heads | 4 |
| | | Linear Projection | $256 \rightarrow 256$ |
| Optimizer | - | Batch Size | 64 |
| | | Maximum Learning Rate | 3e-4 |
| | | Minimum Learning Rate | 5e-5 |
| | | Learning Rate Scheduler | Cosine |
| | | Optimizer Type | AdamW |
| | | Adam $\beta$ | $(0.9, 0.99)$ |
| | | Weight Decay | 0.05 |
| | | Total Epochs | 100 |
| | | Warm-up Epochs | 10 |

Table 9: The hyperparameters for BrainStratify-Fine CLS training.

| Module | Sub-Module | Name | Value |
|---|---|---|---|
| Neural Encoder | - | - | - |
| Label CLS Head | - | Flatten | - |
| | | Linear Projection | $N_f \times 256 \to 128(\text{ReLU}) \to |\mathcal{Y}|$ |
| Optimizer | - | Batch Size | 32 |
| | | Maximum Learning Rate | 2e-4 |
| | | Minimum Learning Rate | 5e-6 |
| | | Learning Rate Scheduler | Cosine |
| | | Optimizer Type | AdamW |
| | | Adam $\beta$ | $(0.9, 0.99)$ |
| | | Weight Decay | 0.05 |
| | | Total Epochs | 200 |
| | | Warm-up Epochs | 20 |

Table 10: The hyperparameters for BrainStratify-Fine RGS training.

| Module | Sub-Module | Name | Value |
|---|---|---|---|
| Neural Encoder | - | - | - |
| Label RGS Head | - | # of Input Channels | {256,256,256,256} |
| | | # of Output Channels | {256,256,256,256} |
| | | Kernel Size | {3,3,3,9} |
| | | Stride | {2,2,2,5} |
| | | Padding | - |
| | | Output Padding | - |
| | | Linear Projection | $256 \to F$ |
| Optimizer | - | Batch Size | 32 |
| | | Maximum Learning Rate | 2e-4 |
| | | Minimum Learning Rate | 5e-6 |
| | | Learning Rate Scheduler | Cosine |
| | | Optimizer Type | AdamW |
| | | Adam $\beta$ | $(0.9, 0.99)$ |
| | | Weight Decay | 0.05 |
| | | Total Epochs | 200 |
| | | Warm-up Epochs | 20 |

Table 11: The hyperparameters for BrainStratify-Fine CTC training.

| Module | Sub-Module | Name | Value |
|---|---|---|---|
| Neural Encoder | - | - | - |
| Label CLS Head | - | Flatten Window | 3 |
| | | Linear Projection | $3 \times 256 \to 128(\text{ReLU}) \to |\mathcal{Y}|$ |
| Optimizer | - | Batch Size | 32 |
| | | Maximum Learning Rate | 2e-4 |
| | | Minimum Learning Rate | 5e-6 |
| | | Learning Rate Scheduler | Cosine |
| | | Optimizer Type | AdamW |
| | | Adam $\beta$ | $(0.9, 0.99)$ |
| | | Weight Decay | 0.05 |
| | | Total Epochs | 200 |
| | | Warm-up Epochs | 20 |

## E   DATA AUGMENTATION

To enhance the robustness of learned representations during both the pre-training and fine-tuning stages, we apply data augmentation in both datasets.

**Pre-training dataset.**   In our implementation, we segment neural recordings into 8-second samples with a 4-second overlap. When fetching a sample, we randomly select a starting point between 0 and 4 seconds, then extract a 4-second sample beginning from that point.

**Downstream dataset.**   Since trials occur consecutively without gaps, employing the jittering mentioned above leads to the blending of information from other trials. In our implementation, we segment sEEG recordings into samples with the corresponding trial length; see Appendix B for details. When fetching a sample, we randomly choose a shift step between 0 and 0.2 seconds, then shift the sample either to the left or right, padding it with zeros.

## F   VISUALIZATION OF SPATIAL CONTEXT CLASSIFICATION

Figure 7 demonstrates the convergence curves of the total pre-training loss and spatial context accuracy of BrainStratify-Coarse. We observe that there is a stable decrease in the spatial context loss, and the spatial context accuracy achieves $\geq 95\%$.

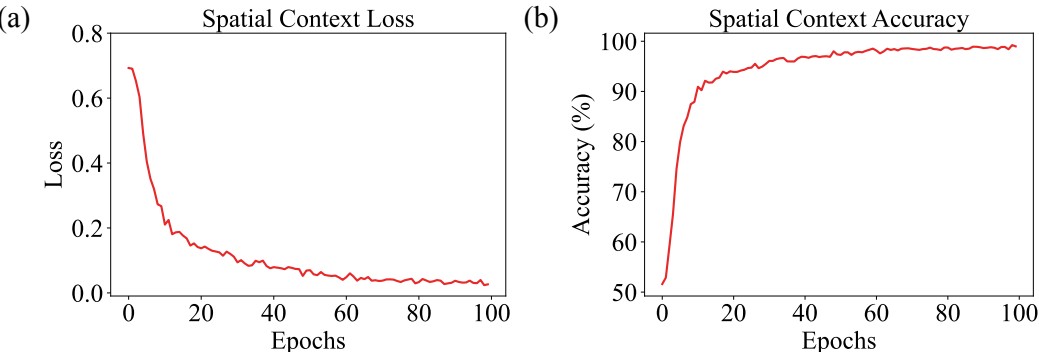

Figure 7: The loss curve and accuracy curve during the training process of BrainStratify-Coarse.

## G   VISUALIZATION OF VECTOR-QUANTIZED NEURAL RECONSTRUCTION

We further visualize how the neural signals are reconstructed. As depicted in Figure 8, although some details are missing, the overall trend of the signals is reconstructed well. Meanwhile, there is a stable decrease in the reconstruction loss during training, which indicates the discrete codex does learn high-level information from neural signals.

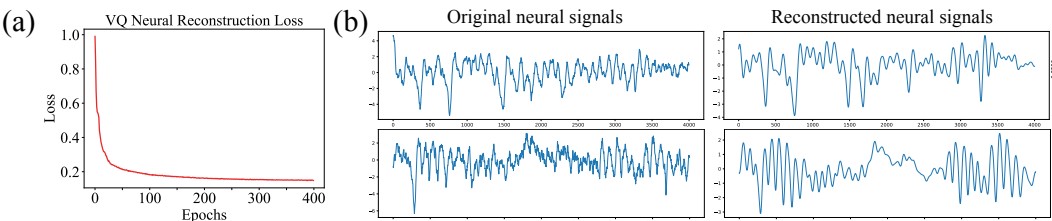

Figure 8: **The visualization of Vector-Quantized Neural Reconstruction. (a).** The reconstruction loss curve during the training process of BrainStratify-Fine VQ-VAE. **(b).** The visualization of reconstructed neural signals.

## H  VISUALIZATION OF MASK NEURAL MODELING

Figure 9 demonstrates the convergence curves of the total pre-training loss and masked neural modeling accuracy of BrainStratify-Fine MAE. We observe that there is a stable decrease in the mask modeling loss, and the mask modeling accuracy achieves about 40%.

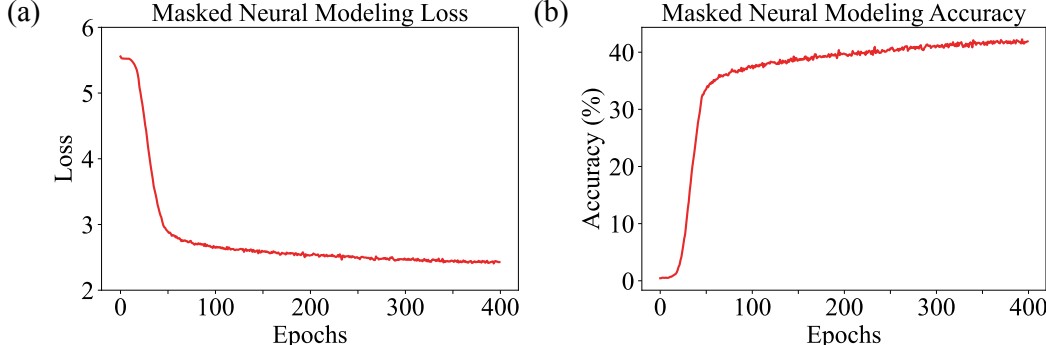

Figure 9: The loss curve and accuracy curve during the training process of BrainStratify-Fine MAE.

## I  WEIGHTED ANATOMICAL COUNTS

To provide quantitative comparisons for the channel cluster, we calculate the weighted anatomy region counts $N_{anatomy}$ wrt. Desikan-Killiany atlas (Desikan et al., 2006) for different channel cluster methods. Specifically, $N_{anatomy} = \sum \frac{C_i}{C} N^i_{anatomy}$, where $C_i$ is the number of channels within that cluster, $C$ is the total number of channels, $N^i_{anatomy}$ is the anatomy region counts in that cluster. A value of $N_{anatomy} = 1$ indicates perfect alignment with anatomical regions, while $N_{anatomy}$ approaching the total number of regions suggests limited functional grouping capability. Thus, lower $N_{anatomy}$ values reflect stronger anatomical alignment in channel clustering.

## J  ADDITIONAL EVALUATION

As demonstrated in Table 12, our method outperforms all advanced neural decoding baselines across multiple decoding paradigms on other datasets, which span data modalities like (epidural) ECoG, MEA, and sEMG, further demonstrating its generalizability across different data modalities.

Table 12: Results on other datasets. Paired T-tests are evaluated between our method and other decoding baselines.

| Methods | Du-IN Spectrogram | Du-IN Wav2Vec2 | Ours Motor | NPTL | emg2qwerty |
|---|---|---|---|---|---|
| Baseline | - | - | - | 80.30±0.13 | 84.62±2.08 |
| LaBraM | 0.7460±0.0121 | 0.7539±0.0046 | 80.39±1.62 | - | - |
| CBraMod | 0.7634±0.0104 | 0.7695±0.0040 | 82.08±0.97 | - | - |
| PopT | 0.7701±0.0107 | 0.7771±0.0039 | 85.94±1.02 | - | - |
| EEG-CFMR | 0.7655±0.0114 | 0.7585±0.0038 | 85.68±1.29 | - | - |
| Du-IN | 0.7807±0.0116 | 0.8127±0.0097 | 88.38±0.96 | 83.11±0.27 | 86.56±1.90 |
| H2DiLR | 0.7439±0.0109 | 0.6768±0.0060 | 77.12±1.84 | 72.68±1.21 | 80.01±2.26 |
| BrainStra.-Fine | **0.7919±0.0121** | **0.8232±0.0100** | **90.65±0.84** | **86.45±0.15** | **88.01±1.75** |

[†] $p < 0.001$ (purple); $p < 0.01$ (pink); $p < 0.05$ (yellow); $p > 0.05$ (gray)

Due to the different temporal resolution of MEA & sEMG recordings (Willett et al., 2023; Sivakumar et al., 2024), we re-optimize the Patch Tokenizer part of Neural Encoder (Table 13 & 14), and share the encoder across all baselines (e.g., Du-IN, H2DiLR). In the NPTL dataset (Willett et al., 2023), we apply a log transformation to smooth the neural features, thus better cooperating with the CNN-based Patch Tokenizer. In the emg2qwerty dataset (Sivakumar et al., 2024), the raw sEMG signals (2kHz) are transformed into spectrogram series (125Hz) before feeding into the baseline model for decoding. Similarly, we directly flatten the spectrogram series $S \in \mathbb{R}^{C \times F \times T}$ into $\mathcal{X} \in \mathbb{R}^{C' \times T}$ with $C' = C \cdot F$, where $C$ is the number of channels, $F$ is the dimension of spectrogram features, $T$ is the total timestamps.

Table 13: The hyperparameters of Neural Encoder for NPTL dataset.

| Module | Name | Value |
|---|---|---|
| | Linear Projection | $C \rightarrow C$ |
| | # of Input Channels | $\{C,\}$ |
| | # of Output Channels | $\{256,\}$ |
| Patch Tokenizer | Kernel Size | $\{9,\}$ |
| | Stride | $\{5,\}$ |
| | Padding | $\{4,\}$ |

Table 14: The hyperparameters of Neural Encoder for the emg2qwerty dataset.

| Module | Name | Value |
|---|---|---|
| | Linear Projection | $C' \rightarrow C'$ |
| | # of Input Channels | $\{C',256,256\}$ |
| | # of Output Channels | $\{256,256,256\}$ |
| Patch Tokenizer | Kernel Size | $\{9,3,3\}$ |
| | Stride | $\{5,1,1\}$ |
| | Padding | $\{4,1,1\}$ |

# K  ADDITIONAL ABLATION STUDY

Leveraging the Du-IN dataset (Zheng et al., 2025) and the word-reading (epidural) ECoG dataset, we conduct thorough ablation studies on model designs to evaluate their effectiveness.

## K.1  ABLATIONS ON CHANNEL CLUSTER

The ablation results on the number of clusters with the sEEG word classification task are provided in Figure 10 (a). Fewer clusters (e.g., 5) reduce the spatial resolution of functional groups and may include irrelevant channels. More clusters maintain performance but require more group-level evaluation and combination.

## K.2  ABLATIONS ON DECOUPLED PRODUCTION QUANTIZATION

We evaluate the hyperparameters of DPQ through comprehensive ablation studies. Figure 10 (b) shows BrainStratify's performance on the sEEG & (epidural) ECoG word classification tasks. We evaluate performance against varying codex groups (from 0 to 16) to ascertain if the number of codex groups affects the quality of the learned codex. To maintain the capacity of the codex with $G = 1$, we set $N_{codex}$ to 2048, while for the other settings, $N_{codex}$ is set to 256. As illustrated in Figure 10 (b.1), since the sEEG channels are pre-selected based on target functional groups, even a small number of codex groups (i.e., $G = 4$ for sEEG instead of $G = 8$ for (epidural) ECoG) can effectively decouple neural components (Silva et al., 2024; Metzger et al., 2023). We also assess performance across different codex sizes (from 64 to 2048) to ascertain if codex size affects the quality of the learned codex. As illustrated in Figure 10 (b.2), while extremely small codex size lacks representation

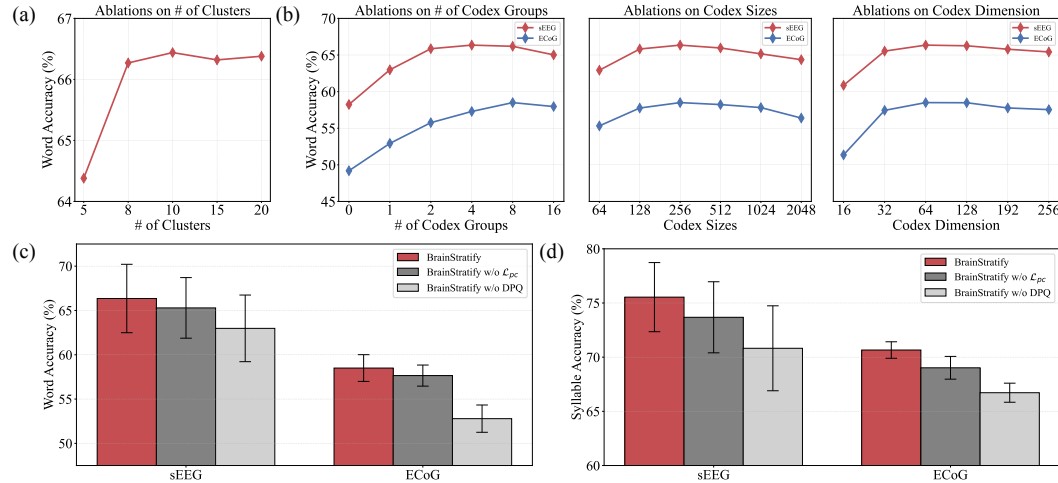

Figure 10: **Ablation studies on model designs. (a).** Ablations on different numbers of clusters with the sEEG word classification task. **(b).** Ablations on different codex groups, codex sizes, and codex dimensions with the sEEG & (epidural) ECoG word classification tasks. **(c) & (d).** Ablations on DPQ sub-modules with sEEG and word-reading (epidural) ECoG datasets (i.e., word classification task and syllable sequential task, respectively).

diversity, extremely large codex size often leads to codex collapse. We suspect that our existing training data might not be adequate for larger codex sizes. Furthermore, our experiments suggest that the model performs best when the dimension of module code $\mathbf{z}_{q[g]}(\boldsymbol{e}_i)$, denoted as $d_{codex} = 64$, is slightly smaller than the model dimension, $d = 256$, resulting in more effective regularization.

To evaluate the contribution of the DPQ components, we perform ablations on the partial-correlation constraint ($\mathcal{L}_{pc}$) and the entire DPQ module with word-reading sEEG and (epidural) ECoG datasets as illustrated in Figure 10 (c) & (d). In implementation, we initialize each sub-quantizer's projection and codex separately, which partially replicates the regularization effect of $\mathcal{L}_{pc}$. Therefore, ablating $\mathcal{L}_{pc}$ alone results in only a modest performance decrease. However, removing DPQ entirely reduces BrainStratify-Fine to Du-IN (Zheng et al., 2025), significantly degrading decoding performance.

## L  MODEL EFFICIENCY

Table 15 shows the FLOPs (with `thop` package) and per-trial inference time across all methods. The total inference time (including data transfer, GPU computation, etc.) is $\leq 20$ ms for all methods, supporting real-time speech decoding. All these results were conducted on 1 NVIDIA 3090 GPU.

Table 15: Model Efficiency Analysis on Du-IN dataset.

| Methods | LaBraM | CBraMod | PopT | EEG-CFMR | Du-IN | H2DiLR | BrainStratify-Fine |
|---|---|---|---|---|---|---|---|
| MFLOPs | 354.48 | 647.43 | 772.69 | 215.48 | **159.69** | 539.07 | 301.06 |
| Time (ms) | 14.07 | 12.17 | 12.59 | **5.79** | 7.60 | 10.69 | 8.94 |

## M  BROADER IMPACTS

BrainStratify has the potential to advance speech decoding and invasive BCI systems by providing a robust and interpretable solution for speech decoding from invasive recordings. Its advanced performance across datasets and modalities while keeping data-efficient makes it well-suited for real-world scenarios where large-scale medical data annotation is often prohibitively costly or unfeasible. Furthermore, our (epidural) ECoG results demonstrate the clinical viability of our method in patients with communication and functional impairments caused by amyotrophic lateral sclerosis (ALS).

# N   SUBJECT-WISE CHANNEL CLUSTER

We provide detailed information on the clusters of the implanted sEEG electrodes for each subject.

## N.1   RESULTS ON DU-IN DATASET

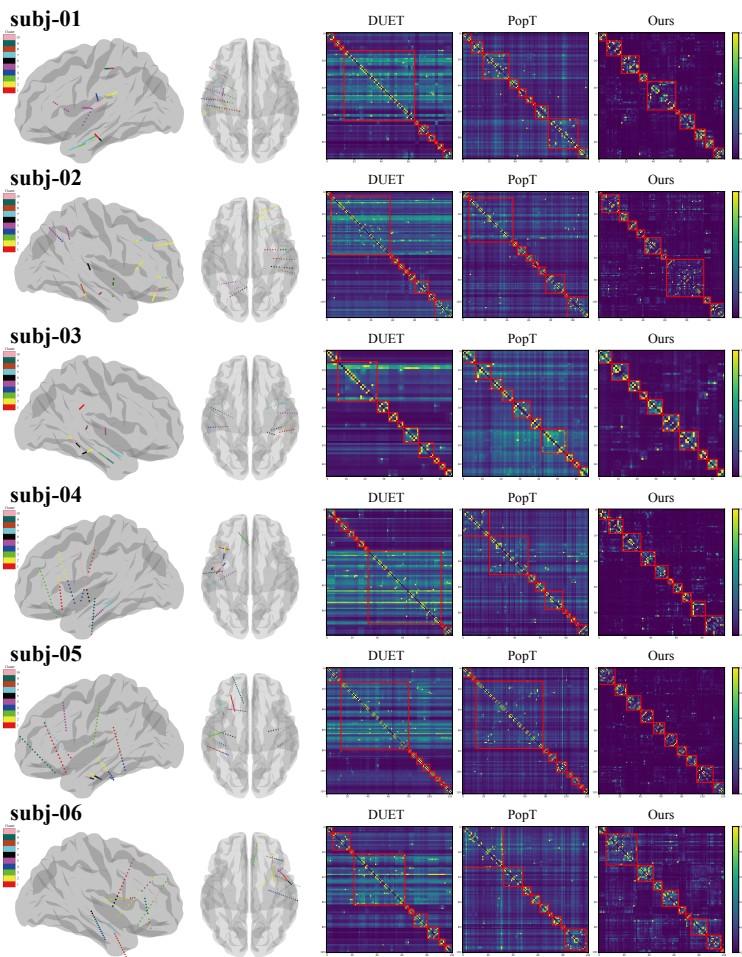

Figure 11: Channel clusters from subjects (01-06).

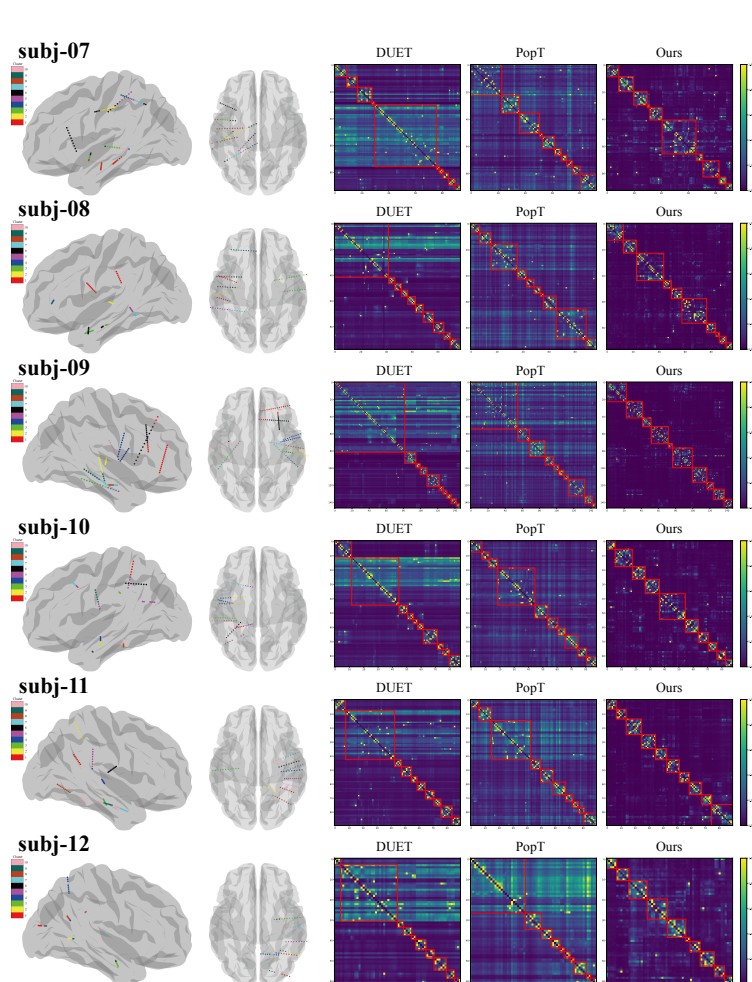

Figure 12: Channel clusters from subjects (07-12).

## N.2 RESULTS ON BRAIN TREEBANK DATASET

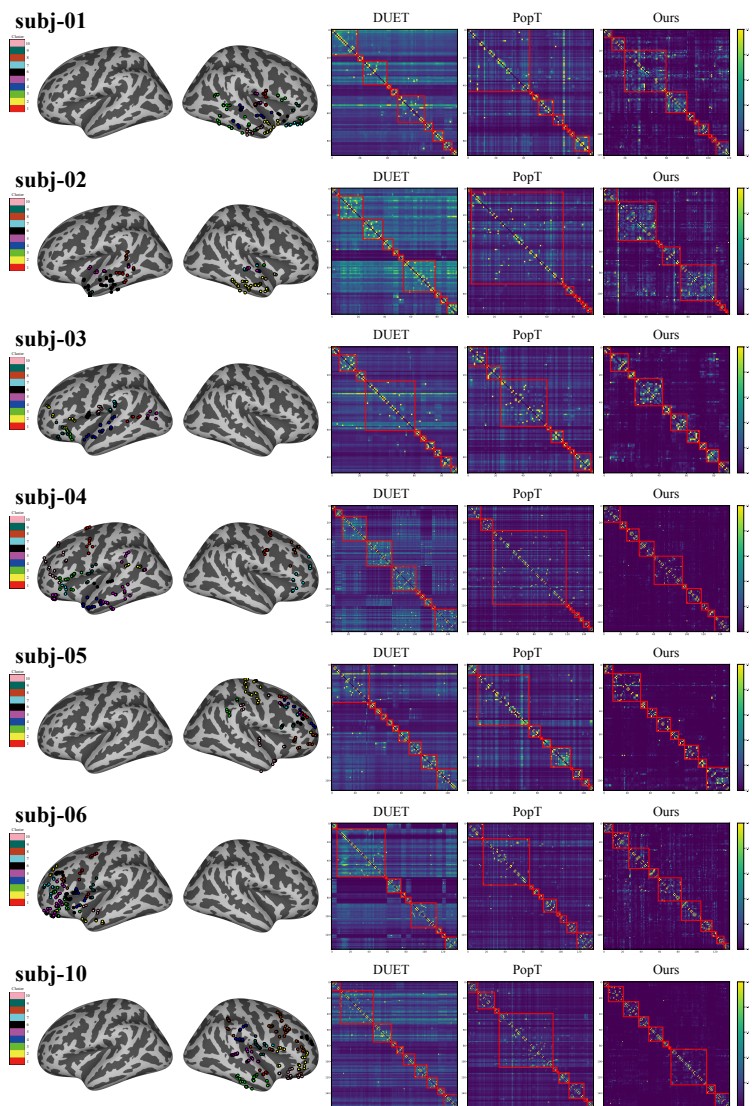

Figure 13: Channel clusters from subjects (01-06,10).

# O    SUBJECT-WISE EVALUATION

## O.1    RESULTS ON DU-IN DATASET

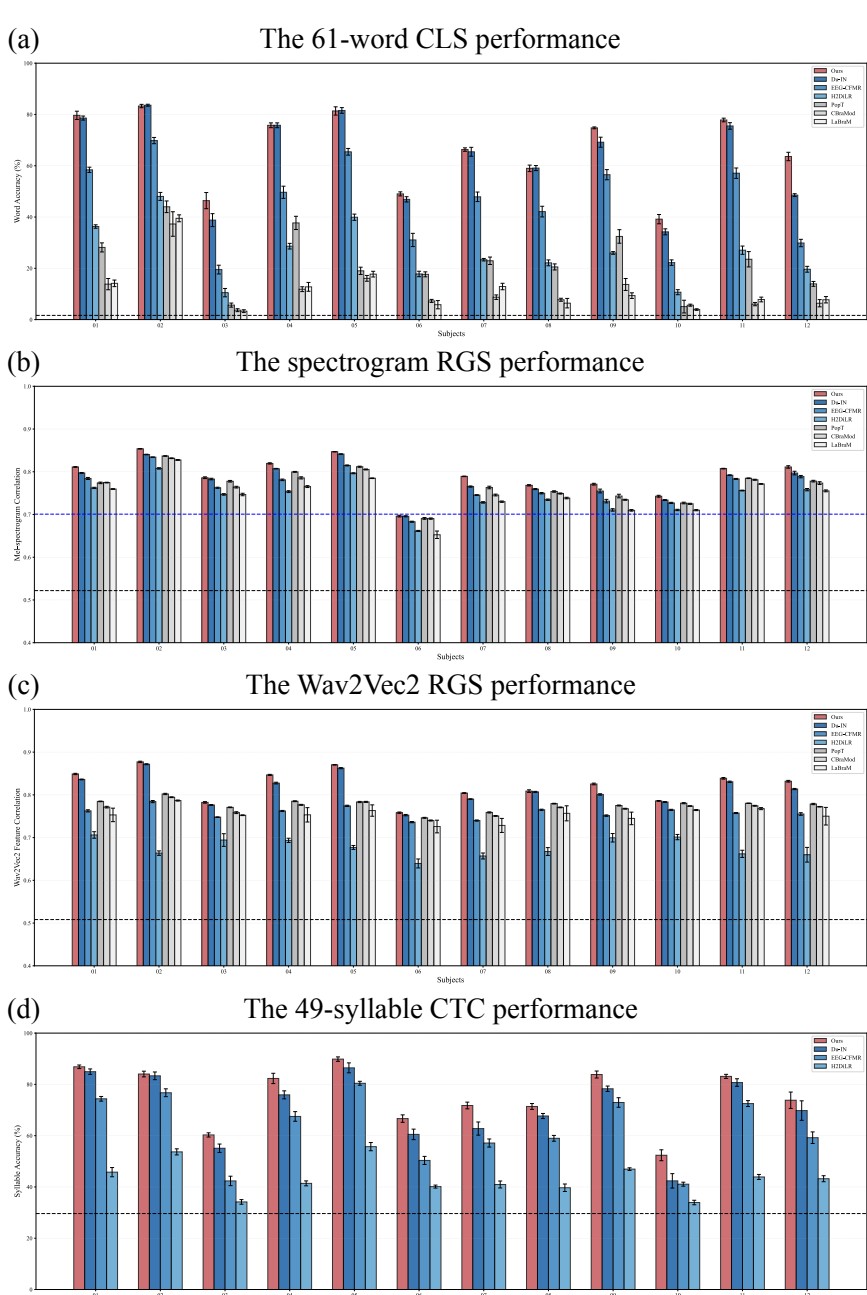

Figure 14: **Model performance on Du-IN dataset. (a).** The 61-word CLS (classification) performance. **(b).** The speech spectrogram RGS (regression) performance. The blue line demonstrates the baseline performance on subject HB02 from Chen et al. (2024). **(c).** The wav2vec 2.0 feature RGS (regression) performance. **(d).** The 49-syllable CTC (connectionist temporal classification) performance. The black line demonstrates the chance-level performance. The error bar demonstrates the standard deviation across 6 random seeds.

## O.2 RESULTS ON BRAIN TREEBANK DATASET

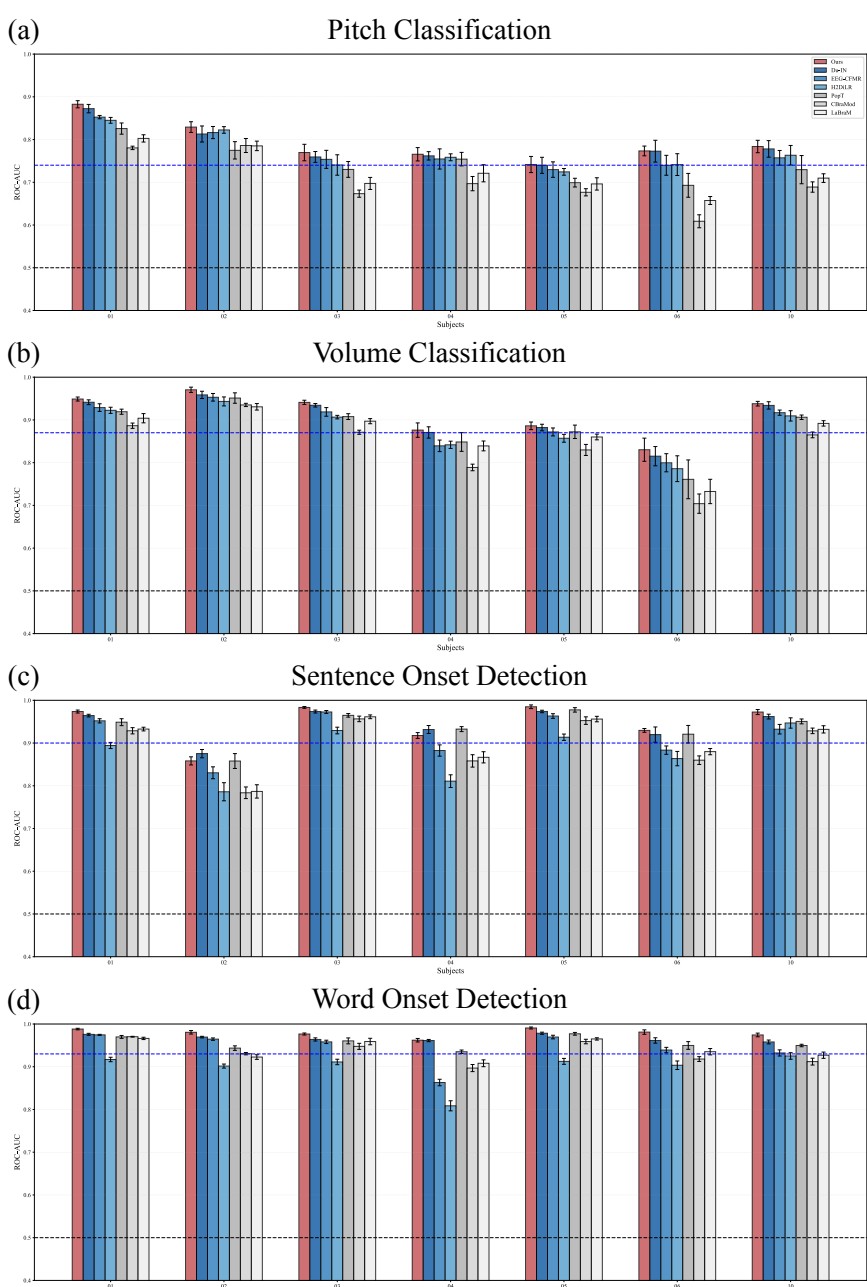

Figure 15: **Model performance on Brain Treebank dataset. (a).** The performance on the pitch classification task. **(b).** The performance on the volume classification task. **(c).** The performance on the sentence onset detection task. **(d).** The performance on the word onset detection task. The black line demonstrates the chance-level performance. The blue line demonstrates the baseline performance from PopT (Chau et al., 2024). The error bar demonstrates the standard deviation across 6 random seeds.

O.3    RESULTS ON EMG2QWERTY DATASET

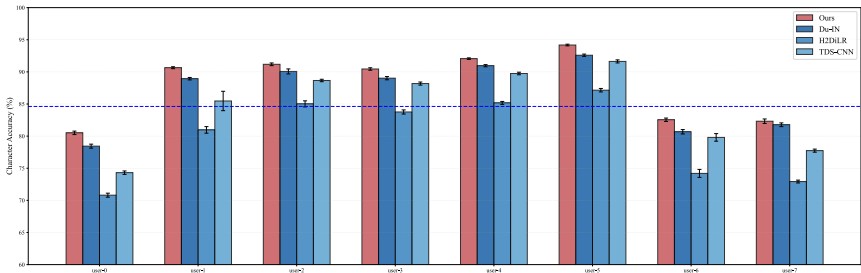

Figure 16: **The 98-character CTC (connectionist temporal classification) performance of different models.** The blue line demonstrates the baseline performance from emg2qwerty (Sivakumar et al., 2024). The error bar demonstrates the standard deviation across 6 random seeds.

