# OpenReview forum: "BrainStratify: Coarse-to-Fine Disentanglement of Intracranial Recordings for Speech Decoding"
_ICLR.cc/2026/Conference — ICLR 2026 Conference Withdrawn Submission_

### Official Review · Reviewer_xusM · 2025-10-27

**Soundness:** 3
**Presentation:** 3
**Contribution:** 2
**Rating:** 4
**Confidence:** 4

**Summary:**

This paper presents BrainStratify, a unified framework for decoding speech from intracranial recordings (sEEG and ECoG) that employs a coarse-to-fine neural disentanglement approach. The framework addresses two key challenges in intracranial neural decoding: (1) task-relevant neural signals are sparsely distributed across electrodes, and (2) multiple neural components (e.g., tongue, jaw, and lips control) are often entangled within functional groups. The proposed solution consists of two stages: (i) Coarse Disentanglement Learning (BrainStratify-Coarse), which identifies functional channel groups through spatial-context-guided temporal-spatial modeling and learns sparse inter-channel attention graphs via self-supervised pre-training, and (ii) Fine Disentanglement Learning (BrainStratify-Fine), which introduces Decoupled Product Quantization (DPQ) to disentangle neural components within target functional groups by using multiple codexes with partial-correlation constraints. The authors evaluate their method on six datasets spanning various modalities (sEEG, epidural ECoG, MEA, sEMG) and tasks (vocal production, speech perception, motor intention), including two newly collected epidural ECoG datasets from patients with ALS and spinal cord injury. Experimental results demonstrate that BrainStratify achieves state-of-the-art performance across all evaluated tasks, with particularly strong improvements on word classification and syllable sequential classification tasks.

The main contributions include: (1) a neuroscience-inspired design that leverages modular brain computation principles to identify functional groups based on learned inter-channel connectivity rather than low-level correlations, (2) a label-efficient approach that requires minimal labeled data for channel selection (600 samples vs. 3000 for supervised methods), (3) a DPQ mechanism that successfully disentangles neural components as validated through articulator control experiments, and (4) comprehensive experimental validation showing consistent improvements over baseline methods including LaBraM, CBraMod, PopT, EEG-CFMR, Du-IN, and H2DiLR. The framework demonstrates strong interpretability through visualizations of learned channel clusters and codex contributions, and achieves real-time inference speeds (≤20ms per trial) suitable for practical BCI applications.

**Strengths:**

The paper presents a genuinely novel approach to intracranial neural decoding by addressing two critical yet under-explored challenges simultaneously: sparse electrode distribution and entangled neural components. The originality lies in the elegant coarse-to-fine disentanglement framework that mirrors hierarchical brain organization principles. Unlike existing methods (CCM, DUET) that rely on low-level correlation-based clustering—which the authors convincingly demonstrate to be inadequate for multimodal intracranial signals where similar firing patterns hold different functional meanings by location—BrainStratify learns functional groupings through spatial-context-guided self-supervised pre-training. The introduction of Decoupled Product Quantization (DPQ) for neural component disentanglement is particularly innovative, adapting product quantization from computer vision to the neuroscience domain with partial-correlation constraints that encourage codex independence. The articulator control validation experiments (Figure 4) provide compelling evidence that learned codexes capture interpretable neural components (jaw, lips, tongue control), demonstrating that the method discovers meaningful neuroscientific structure rather than arbitrary feature decompositions.

The experimental evaluation demonstrates outstanding thoroughness across multiple dimensions. The evaluation spans six diverse datasets covering different modalities (sEEG, epidural ECoG, MEA, sEMG), recording scenarios (vocal production, speech perception, motor intention), and subject populations (12 subjects in Du-IN, 10 in Brain Treebank, plus newly collected clinical patients), providing strong evidence of generalizability. The authors collect two new well-annotated clinical datasets from patients with ALS and spinal cord injury, contributing valuable resources to the community. The statistical analysis is rigorous, reporting results with standard errors across six random seeds and conducting paired t-tests against all baselines. The ablation studies are comprehensive, systematically evaluating the impact of cluster numbers, codex groups, codex sizes, codex dimensions, and individual DPQ components. The label-efficiency analysis (Figure 1) demonstrates a critical practical advantage: while supervised methods require 3000 samples to identify task-relevant channels, BrainStratify-Coarse achieves comparable performance with only 600 samples, addressing a fundamental constraint in medical data collection where large-scale annotation is often prohibitively expensive.

The paper is exceptionally well-written with clear motivation, logical flow, and comprehensive documentation. The problem formulation is precise, clearly distinguishing between coarse-grained functional group identification and fine-grained neural component disentanglement. Figure 2 provides an excellent overview of the entire framework, effectively illustrating both training stages and downstream applications. The visualization of channel connectivity matrices (Figure 3b) offers direct comparison with baselines, making the superiority of the learned sparse attention graphs immediately apparent. The anatomical visualizations (Figure 3c-d) ground the learned clusters in neuroscientific interpretability by showing weighted anatomical counts relative to the Desikan-Killiany atlas. The disentanglement evaluation (Figure 4) is particularly compelling, showing channel tuning maps across articulators and quantifying each codex's contribution to different articulator control tasks. The extensive appendices provide complete implementation details, hyperparameters, subject-wise visualizations, and convergence curves, ensuring full reproducibility.

The work makes important contributions toward clinically deployable speech BCIs for patients with communication impairments. The state-of-the-art results across all evaluated tasks demonstrate consistent and meaningful performance improvements over strong baselines including Du-IN, PopT, and other foundation models. The framework's data efficiency (requiring only 600 samples for effective channel selection) and fast inference speed (≤20ms per trial) make it suitable for real-world deployment where calibration time and response latency are crucial. The successful application to epidural ECoG—a minimally invasive recording modality positioned outside the dura mater that reduces tissue damage compared to traditional intracranial placements—is particularly noteworthy for clinical translation. The results on patients with severe motor impairments (ALS patient who lost communication abilities, SCI patient with complete limb paralysis) demonstrate real-world applicability. The interpretability provided by learned functional groups and disentangled neural components enhances clinical trust and enables neuroscientists to validate and refine the system based on known brain organization principles.

**Weaknesses:**

While the paper presents a comprehensive framework, several theoretical and methodological concerns warrant deeper investigation. First, the theoretical justification for why the spatial context pre-training task (replacing 10\% of channels with unrelated temporal activity) should lead to learning functional connectivity remains somewhat underspecified. The authors claim this avoids shortcuts in mask-based reconstruction where models might over-rely on intra-channel temporal patterns, but the connection between detecting temporal mismatches and learning inter-channel functional relationships is not rigorously established. The rapid convergence to ≥95\% accuracy with only 1-hour data suggests the task may be relatively superficial, detecting obvious temporal discontinuities rather than learning deep functional relationships. A more thorough analysis comparing the learned attention patterns with established neuroscientific measures of functional connectivity (e.g., coherence, Granger causality, phase-locking value) would strengthen the claim that BrainStratify-Coarse genuinely captures functional groupings rather than simpler statistical dependencies. Additionally, the paper lacks theoretical analysis of what properties the learned channel connectivity matrix P should possess to ensure meaningful clustering, and there is no discussion of convergence guarantees or sensitivity to initialization.

The experimental setup and comparison framework present several limitations that may overstate the improvements. Most critically, the comparison with baseline methods is not entirely fair in terms of architectural choices and pre-training strategies. For instance, the authors pre-train PopT across all subjects in a dataset but pre-train BrainStratify-Coarse on individual subjects, making it difficult to disentangle whether improvements stem from the proposed method or from subject-specific tuning. The paper claims PopT "struggles to converge with limited data" and thus requires multi-subject pre-training, but this asymmetric setup raises questions about whether PopT would perform better with subject-specific pre-training given sufficient data augmentation or optimization adjustments. Similarly, for DUET and CCM, the authors use raw (non-rereferenced) signals while using rereferenced signals for BrainStratify, justified by claiming rereferencing "disrupts correlations" needed by correlation-based methods. However, this inconsistency means the methods are operating on fundamentally different input representations, making direct performance comparison problematic. A more rigorous evaluation would either (1) adapt all methods to work with the same input representation, or (2) provide extensive ablations showing that each method performs best with its chosen representation. The paper also lacks comparison with more recent neural decoding methods or domain adaptation techniques that might handle cross-subject variability more effectively.

The generalization and scalability claims require more careful scrutiny. While the paper evaluates on six datasets, several critical limitations constrain the generalizability conclusions. First, five of the six datasets involve speech-related tasks (vocal production, speech perception, reading), with only one motor intention dataset, limiting evidence that the method generalizes beyond the speech domain. The authors acknowledge this limitation, stating the framework's "design affords clear interpretability grounded in brain organization" and anticipating generalization to other cognitive states, but this remains speculative without empirical validation on diverse cognitive tasks like visual perception, memory, decision-making, or emotional processing. Second, the epidural ECoG datasets contain only single subjects each, making it impossible to assess inter-subject variability for this clinically important modality. Third, all sEEG/ECoG datasets involve relatively sparse electrode coverage (averaging ~110-128 channels), and it remains unclear how the method would scale to higher-density recordings (e.g., 256+ channels) or whether the identified functional groups would remain stable. The authors note that BrainStratify is "currently limited to identifying functional modules rather than full functional networks" due to sparse electrode distribution, but provide no analysis of how coverage density affects performance or what minimum coverage is required for reliable functional group identification. Fourth, the cross-dataset evaluation is limited, models are trained and tested within the same dataset rather than evaluating transfer across datasets, which would provide stronger evidence of learning generalizable neural representations.

The Decoupled Product Quantization component, while innovative, raises several conceptual and practical concerns. The choice of using G=4 codex groups for sEEG versus G=8 for ECoG is justified post-hoc through ablation studies (Figure 10b), but the paper provides no principled method for determining the optimal number of codex groups a priori for new datasets or modalities. The ablation shows performance is relatively stable across a range of values (4-16 for ECoG), but this sensitivity analysis does not address the fundamental question of how many true neural components should exist within a functional group, a question that likely depends on the specific brain region, task demands, and recording modality. The partial correlation constraint L_pc encourages independence between codexes, but the paper does not analyze whether the learned codexes satisfy independence statistically (e.g., through mutual information analysis) or whether independence is the appropriate inductive bias. In neuroscience, neural components may exhibit structured dependencies (e.g., coordinated jaw-tongue movements during speech), and enforcing strict independence might discard meaningful covariance structure. The disentanglement evaluation (Figure 4) on articulator control tasks is compelling but limited in scope, it shows that different codexes contribute differentially to jaw/lips/tongue control, but does not rigorously quantify disentanglement quality using established metrics from the disentanglement learning literature (e.g., MIG, SAP, DCI scores). Moreover, this evaluation uses only ~2k trials with 6 movement types from a single subject, and it is unclear whether the discovered decomposition would remain consistent across subjects or tasks.

The paper's treatment of related work and positioning within the broader literature could be strengthened. The discussion of disentanglement representation learning (Section 2.3) mentions several relevant methods (QLAE, Tripod, pi-VAE, PDisVAE) but does not empirically compare against any of them, making it difficult to assess whether DPQ offers advantages over existing disentanglement techniques when applied to neural data. The claim that "neural signals require disentanglement along the channel dimension, similar to how RGB channels are treated in images" is conceptually interesting but somewhat misleading, as RGB channels represent different wavelengths of the same spatial location, while neural channels represent different spatial locations with potentially different functional properties. This distinction matters for justifying architectural choices. Additionally, the paper does not adequately discuss recent work on multimodal disentanglement or structured latent variable models that might be applicable. The comparison with time series forecasting methods (CCM, DUET) is somewhat tangential since these methods optimize for prediction rather than decoding, and their failure on neural decoding tasks does not necessarily invalidate their core ideas about channel clustering. A more relevant comparison would be with other neural decoding methods that explicitly model channel interactions or functional connectivity.

Several implementation details and design choices lack sufficient justification. The paper uses spectral clustering with default scikit-learn parameters for grouping channels, but provides no analysis of robustness to this choice—would other clustering algorithms (e.g., hierarchical clustering with different linkage criteria, community detection algorithms, Louvain modularity optimization) yield similar functional groups? The choice of k=10 clusters appears arbitrary, justified only through post-hoc ablation showing performance is relatively stable across 5-20 clusters, but there is no principled method for selecting k or determining whether the true number of functional groups varies by subject. The weighted anatomical counts metric (Appendix I) is introduced as a quantitative measure of cluster quality, but its interpretation is unclear. why should lower N_anatomy indicate better clustering, and what is the expected value under a null model? The paper reports that "most models surpass the PopT baseline with SC selection" when evaluated on channels selected by BrainStratify-Coarse, but this comparison conflates channel selection quality with model performance, and it is unclear whether improvements stem from better channel selection or from the specific channels selected happening to be easier for certain model architectures. The training computational cost is mentioned briefly (6 hours on V100 GPU), but there is no analysis of how this scales with dataset size, number of channels, or number of subjects, which is relevant for practical deployment.

Finally, the clinical validation and real-world applicability claims require more cautious interpretation. While the results on ALS and SCI patients are promising, these are single-subject cases with relatively short-term evaluation, and the paper does not address critical questions for clinical deployment such as long-term stability of the learned functional groups across days or weeks, robustness to electrode impedance changes, or performance degradation in the presence of neural plasticity or disease progression. The claim that epidural ECoG is "minimally invasive" and "reduces tissue damage" compared to traditional placements is important for clinical translation, but the paper does not discuss other critical clinical considerations such as infection risk, hardware reliability, or patient acceptance. The fast inference speed (≤20ms) is promising for real-time BCI, but the paper does not analyze latency breakdown (data transfer vs. computation) or discuss whether this latency is acceptable for natural communication, which typically requires <100ms speech decoding latency for fluid conversation. The lack of online closed-loop evaluation. where the BCI system provides real-time feedback to users—limits conclusions about practical usability, as offline decoding performance often overestimates online performance due to non-stationarities and adaptation effects.

**Questions:**

Please see your weaknesses and I will adjust the final score based on your answers.

---

> ### Author Response · Authors · 2025-11-27
> **Response to Reviewer xusM by Authors (1/8)**
>
> Thank you for the thorough review and constructive comments. We are deeply grateful for acknowledging that our method "presents a genuinely novel approach to intracranial neural decoding by addressing two critical yet under-explored challenges simultaneously" and “demonstrates outstanding thoroughness across multiple dimensions". Please see below for the point-by-point responses to your comments. All modifications arising from the rebuttal stage will be incorporated into both the final paper and the GitHub repository.
>
> **Answer to W1**:
>
> > ...First, the theoretical justification for why the spatial context pre-training task (replacing 10% of channels with unrelated temporal activity) should lead to learning functional connectivity remains somewhat underspecified...detecting obvious temporal discontinuities rather than learning deep functional relationships.
>
> Thank you for the valuable feedback. To clarify, the replaced segments are sampled from different time points (could be different sessions) but the same channel. Since all sEEG signals are z-score normalized per channel during preprocessing, the replaced segments have low-level statistics (zero mean and unit variance) similar to those of the original data. This design is intentional to eliminate simple intra-channel statistical cues (e.g., mean or variance shifts) as shortcuts. Instead, the model must rely on the multivariate functional context -- i.e., whether the activity of a channel is coherent with that of all other channels in the current sample. **By replacing a channel with data from a different temporal context (same channel, different time points), we force the model to learn inter-channel dependencies and functional relationships, which are essential for robust functional connectivity estimation.**
>
> > A more thorough analysis comparing the learned attention patterns with established neuroscientific measures of functional connectivity (e.g., coherence, Granger causality, phase-locking value) would strengthen the claim that BrainStratify-Coarse genuinely captures functional groupings rather than simpler statistical dependencies.
>
> Thank you for the helpful suggestion. To empirically validate this advantage, we performed a comparative analysis where channels were clustered using coherence-based functional connectivity (FC) and selected channel groups based on downstream task performance. On the Du-IN dataset, this method achieved an accuracy of 42.04$\pm$3.68 (%), which is modestly lower than the performance achieved with our DUET-based FC (Figure 3a).
>
> We note that this performance gap is relatively small, which is expected. **Since both DUET and the coherence-based method are fundamentally derived from correlation metrics, they capture similar, albeit not identical, aspects of channel relationships. The key advantage of our method lies in its more nuanced, context-aware estimation of functional modules, as visualized in Figure 3.**
>
> > Additionally, the paper lacks theoretical analysis of what properties the learned channel connectivity matrix P should possess to ensure meaningful clustering, and there is no discussion of convergence guarantees or sensitivity to initialization.
>
> Thank you for the helpful suggestion. The above results confirm a key limitation of **coherence-based FC: its reliance on low-order correlations makes it susceptible to anatomical proximity, preventing it from delineating precise functional boundaries as effectively as our method**. In contrast, our approach infers functional organization directly from the model's spatial attention, producing a more accurate and functionally relevant grouping that explains the performance differential.
>
> **Our clustering objective is guided by the neuroscientific principle of modular brain organization [1,2], which favors sparse, robust connectivity matrices.** The BrainStratify-Coarse architecture (Section 3.2) was carefully designed with this in mind to best estimate channel dependencies. While a formal convergence analysis is beyond the decoding focus of this paper, our method demonstrated stable convergence across all subjects in available sEEG datasets. Please see Appendix N for comprehensive subject-wise connectivity results.
>
> **References**:
>
> [1] Buzsáki G. Rhythms of the Brain[M]. Oxford university press, 2006.
>
> [2] Silva A B, Littlejohn K T, Liu J R, et al. The speech neuroprosthesis[J]. Nature Reviews Neuroscience, 2024, 25(7): 473-492.

---

> > ### Author Response · Authors · 2025-11-27
> > **Response to Reviewer xusM by Authors (2/8)**
> >
> > **Answer to W2**:
> >
> > > ...Most critically, the comparison with baseline methods is not entirely fair in terms of architectural choices and pre-training strategies. For instance, the authors pre-train PopT across all subjects in a dataset but pre-train BrainStratify-Coarse on individual subjects, making it difficult to disentangle whether improvements stem from the proposed method or from subject-specific tuning...optimization adjustments.
> >
> > Thank you for the helpful suggestion. We also evaluated PopT pre-trained on a single subject, using its estimated channel connectivity for clustering. This approach achieved 47.08$\pm$3.96 (%) on the Du-IN dataset, which is modestly lower than the performance achieved when PopT was pre-trained on all subjects. We found that, **during pre-training, only 4 out of 12 single-subject models converged to the target ~85% performance, indicating a sensitivity to data volume**.
> >
> > To directly address PopT's high data requirement, we modified the loss function to more explicitly contrast embeddings (Line 207). This adjustment is designed to improve learning efficiency when data is limited. We highlight a key distinction: while PopT similarly uses spatial context discrimination, its goal is to build a general foundation model for neural decoding. In contrast, our work **retargets this architecture by specifically demonstrating its power for enabling highly data-efficient channel selection, and we provide robust neuroscientific interpretability through visualization**.
> >
> > > Similarly, for DUET and CCM, the authors use raw (non-rereferenced) signals while using rereferenced signals for BrainStratify, justified by claiming rereferencing "disrupts correlations" needed by correlation-based methods. However, this inconsistency means the methods are operating on fundamentally different input representations, making direct performance comparison problematic...more effectively.
> >
> > Thank you for the helpful suggestion. When evaluated on rereferenced data, CCM and DUET achieved performances of 29.42$\pm$5.13 (%) and 35.70$\pm$4.59 (%), respectively, on the Du-IN dataset. This suboptimal performance stems from a fundamental limitation: like coherence-based FC, these methods rely on low-order signal correlations, which are inherently linked to anatomical proximity. However, a critical issue arises after **rereferencing -- this preprocessing step effectively removes the anatomical cues from the signal**. Consequently, without explicit spatial coordinates, the models cannot recover the underlying anatomy and thus fail to accurately estimate functional boundaries, leading to poor channel selection.

---

> > > ### Author Response · Authors · 2025-11-27
> > > **Response to Reviewer xusM by Authors (3/8)**
> > >
> > > **Answer to W3**:
> > >
> > > > ...While the paper evaluates on six datasets, several critical limitations constrain the generalizability conclusions. First, five of the six datasets involve speech-related tasks (vocal production, speech perception, reading), with only one motor intention dataset, limiting evidence that the method generalizes beyond the speech domain...emotional processing.
> > >
> > > Thank you for the insightful feedback. We have limited our current claims to speech decoding due to the scope of publicly available sEEG benchmarks, which are predominantly in this domain. This allows for a direct and rigorous comparison with existing state-of-the-art methods. However, it is important to clarify that our framework is fundamentally designed around general neuroscience principles of modular computation and functional connectivity. The architecture is not intrinsically tied to speech. Therefore, we fully anticipate that its core methodology will generalize effectively to other cognitive tasks as suitable datasets become available. Following, we want to clarify the rationale for limiting our claims to speech decoding.
> > >
> > > First, it’s worth noting that, due to the lack of publicly available intracranial datasets, **our evaluation primarily focuses on speech decoding tasks (e.g., vocal production and speech perception)**, which constitute 4 of our 6 benchmark tasks.
> > >
> > > Second, the core design of BrainStratify-Coarse is inspired by the brain's modular organization. **This design principle is largely consistent with research on the neural coding of cognitive function organization, which aims to break down cognitive function and identify the stable and precise information encoded by each brain region (such as vSMC encoding speech motor and STG encoding auditory perception).** For such cognitive-decoding tasks, our method aims to estimate the precise functional boundaries of these modules, as demonstrated in Figure 3, thus avoiding the involvement of irrelevant channels to reduce performance (Du-IN’s Figure 4). **However, decoding tasks are not limited to cognitive functions; there are other types of decoding tasks, such as epilepsy detection, Alzheimer's disease detection, and sleep staging. For these types of tasks, SOTA decoding models are primarily based on the neurofoundation models [1,2].** Therefore, our model is not capable of handling all types of decoding tasks, but focuses on decoding tasks related to cognitive functions (e.g., speech decoding). Besides, our method can potentially help exploratory neural cognitive-decoding tasks, whose involved brain regions are unclear, as discussed in “General Rebuttal by Authors”.
> > >
> > > Third, to better understand the disentanglement of BrainStratify-Fine, we need to collect datasets with the simplified task space (Section 4.5) according to previous neural encoding studies [3]. **Due to the lack of more public datasets suitable for evaluating disentanglement, we presently confine our claim to the speech decoding field, maintaining a rigorous and focused claim.** That said, we agree that generalization to other tasks is key. We preliminarily evaluated our framework on an ECoG motor-intention task and an EMG-based typing task. The positive decoding (not disentanglement) results on these non-language tasks suggest promising potential for broader extrapolation, and non-language tasks are worth systematically exploring if the corresponding datasets for studying encoding mechanisms are available.
> > >
> > > > Second, the epidural ECoG datasets contain only single subjects each, making it impossible to assess inter-subject variability for this clinically important modality.
> > >
> > > Thank you for the valuable feedback. **In clinical settings, these patients are relatively special and rare, such as ALS patients with long-term implanted electrodes. Most related studies [3-5] have only 1\~2 subjects.** Due to the limitations of open-source datasets, subject-specific modeling and evaluation are acceptable based on invasive recordings (e.g., [6] only includes 2 subjects). Besides, previous studies [7,8] typically evaluate neural disentanglement results with real-world neural spike recordings collected from 1~2 NHPs. We additionally provide the disentanglement results on our collected motor intention dataset as follows to further demonstrate the effectiveness. Specifically, we evaluate the three 2-way classification tasks of right-[hand,elbow,leg], with left-finger as the other state, and report the absolute accuracy improvement against the chance level (50%) for better illustration. The codex-wise contributions of right-[hand,elbow] and right-leg vary, demonstrating that DPQ can distinguish motor intentions with large differences.

---

> > > > ### Author Response · Authors · 2025-11-27
> > > > **Response to Reviewer xusM by Authors (4/8)**
> > > >
> > > > | Codex | 1 | 2 | 3 | 4 | 5 | 6 | 7 | 8 |
> > > > | --- | --- | --- | --- | --- | --- | --- | --- | --- |
> > > > | Right-Hand | 4.55$\pm$1.22 | **22.71$\pm$2.91** | 6.94$\pm$2.19 | **18.72$\pm$1.59** | 8.79$\pm$1.68 | 5.19$\pm$2.01 | 6.08$\pm$1.12 | **14.19$\pm$1.29** |
> > > > | Right-Elbow | 2.10$\pm$0.79 | **20.09$\pm$1.75** | 7.16$\pm$1.22 | **15.92$\pm$0.94** | 6.73$\pm$1.44 | 6.11$\pm$0.92 | 5.70$\pm$1.32 | **17.52$\pm$2.19** |
> > > > | Right-Leg | 1.29$\pm$0.52 | **25.91$\pm$1.68** | **20.16$\pm$1.66** | 8.54$\pm$2.41 | 3.16$\pm$1.19 | 9.17$\pm$2.04 | 7.26$\pm$1.41 | 5.70$\pm$1.81 |
> > > >
> > > > > Third, all sEEG/ECoG datasets involve relatively sparse electrode coverage (averaging ~110-128 channels), and it remains unclear how the method would scale to higher-density recordings (e.g., 256+ channels) or whether the identified functional groups would remain stable...group identification.
> > > >
> > > > Thank you for the valuable feedback. **Our framework is designed for clinical sEEG settings, where the number of implanted electrodes typically ranges from `50` to `200`.** Our experimental results demonstrate robust functional group identification across this entire range of clinical configurations. Extending our framework to infer dynamic network topology would enrich its neuroscientific insight if denser sEEG electrode coverages become available and models are refined to capture time-lagged spatial relationships.
> > > >
> > > > > Fourth, the cross-dataset evaluation is limited, models are trained and tested within the same dataset rather than evaluating transfer across datasets, which would provide stronger evidence of learning generalizable neural representations.
> > > >
> > > > Thank you for the valuable feedback. **Cross-subject transfer is a challenging problem in sEEG modeling, and this is not the core aim of our study. Almost all previous studies [9-12] based on sEEG perform subject-specific fine-tuning.** In clinical settings, different patients may have electrodes implanted in different brain areas with little overlap, and the course of ALS disease is also inconsistent among patients. Therefore, current studies based on intracranial recordings primarily focus on improving the practicability of BCI systems within a single subject (i.e., developing a general framework to enhance decoding performance across different decoding tasks).
> > > >
> > > > **References**:
> > > >
> > > > [1] Yuan Z, Shen F, Li M, et al. Brainwave: A brain signal foundation model for clinical applications[J]. arXiv preprint arXiv:2402.10251, 2024.
> > > >
> > > > [2] Jiang W B, Zhao L M, Lu B L. Large brain model for learning generic representations with tremendous EEG data in BCI[J]. arXiv preprint arXiv:2405.18765, 2024.
> > > >
> > > > [3] Metzger S L, Littlejohn K T, Silva A B, et al. A high-performance neuroprosthesis for speech decoding and avatar control[J]. Nature, 2023, 620(7976): 1037-1046.
> > > >
> > > > [4] Willett F R, Kunz E M, Fan C, et al. A high-performance speech neuroprosthesis[J]. Nature, 2023, 620(7976): 1031-1036.
> > > >
> > > > [5] Moses D A, Metzger S L, Liu J R, et al. Neuroprosthesis for decoding speech in a paralyzed person with anarthria[J]. New England Journal of Medicine, 2021, 385(3): 217-227.
> > > >
> > > > [6] Cho C J, Chang E, Anumanchipalli G. Neural latent aligner: cross-trial alignment for learning representations of complex, naturalistic neural data[C]//International Conference on Machine Learning. PMLR, 2023: 5661-5676.
> > > >
> > > > [7] Liu R, Azabou M, Dabagia M, et al. Drop, swap, and generate: A self-supervised approach for generating neural activity[J]. Advances in neural information processing systems, 2021, 34: 10587-10599.
> > > >
> > > > [8] Wang Y, Li C, Li W, et al. Exploring behavior-relevant and disentangled neural dynamics with generative diffusion models[J]. Advances in Neural Information Processing Systems, 2024, 37: 34712-34736.
> > > >
> > > > [9] Chau G, Wang C, Talukder S, et al. Population transformer: Learning population-level representations of neural activity[J]. ArXiv, 2025: arXiv: 2406.03044 v4.
> > > >
> > > > [10] Zheng H, Wang H, Jiang W, et al. Du-IN: Discrete units-guided mask modeling for decoding speech from Intracranial Neural signals[J]. Advances in Neural Information Processing Systems, 2024, 37: 79996-80033.
> > > >
> > > > [11] Wu D, Li S, Feng C, et al. Towards Homogeneous Lexical Tone Decoding from Heterogeneous Intracranial Recordings[C]//The Thirteenth International Conference on Learning Representations.
> > > >
> > > > [12] Mentzelopoulos G, Chatzipantazis E, Ramayya A G, et al. Neural decoding from stereotactic EEG: accounting for electrode variability across subjects[J]. Advances in Neural Information Processing Systems, 2024, 37: 108600-108624.

---

> > > > > ### Author Response · Authors · 2025-11-27
> > > > > **Response to Reviewer xusM by Authors (5/8)**
> > > > >
> > > > > **Answer to W4**:
> > > > >
> > > > > > ...The choice of using G=4 codex groups for sEEG versus G=8 for ECoG is justified post-hoc through ablation studies (Figure 10b), but the paper provides no principled method for determining the optimal number of codex groups a priori for new datasets or modalities...recording modality.
> > > > >
> > > > > Thank you for the insightful feedback. **Determining the number of potential components contained in a recorded signal is an unsolved and difficult problem, requiring consideration of multiple factors such as the spatial resolution of the recorded signal and the size of the brain region it covers.** Generally, this hyperparameter is selected based on experimental results. However, in practice, it usually needs to be slightly larger than the estimated number of target components [1].
> > > > >
> > > > > We acknowledge that this is a very interesting and important question, and there seems to be some work [2] exploring this area, but our work mainly focuses on the BCI field.
> > > > >
> > > > > > The partial correlation constraint L_pc encourages independence between codexes...this evaluation uses only ~2k trials with 6 movement types from a single subject, and it is unclear whether the discovered decomposition would remain consistent across subjects or tasks.
> > > > >
> > > > > Thank you for the helpful suggestion. We strictly follow the evaluation protocol commonly adopted in the representation learning based neural disentanglement field [7] (Line 427). In clinical settings, these patients are relatively special and rare, such as ALS patients with long-term implanted electrodes. Most related studies [3-5] have only 1\~2 subjects. Due to the limitations of open-source datasets, subject-specific modeling and evaluation are acceptable based on invasive recordings (e.g., [6] only includes 2 subjects). Besides, previous studies [7,8] typically evaluate neural disentanglement results with real-world neural spike recordings collected from 1~2 NHPs. We additionally provide the disentanglement results on our collected motor intention dataset in **Answer to W3**.
> > > > >
> > > > > **References**:
> > > > >
> > > > > [1] Li C, Wang Y, Wang Y, et al. A Revisit of Total Correlation in Disentangled Variational Auto-Encoder with Partial Disentanglement[J]. arXiv preprint arXiv:2502.02279, 2025.
> > > > >
> > > > > [2] Anonymous et al. Estimating Dimensionality of Neural Representations from Finite Samples. https://openreview.net/forum?id=iM4o9a83F7
> > > > >
> > > > > [3] Metzger S L, Littlejohn K T, Silva A B, et al. A high-performance neuroprosthesis for speech decoding and avatar control[J]. Nature, 2023, 620(7976): 1037-1046.
> > > > >
> > > > > [4] Willett F R, Kunz E M, Fan C, et al. A high-performance speech neuroprosthesis[J]. Nature, 2023, 620(7976): 1031-1036.
> > > > >
> > > > > [5] Moses D A, Metzger S L, Liu J R, et al. Neuroprosthesis for decoding speech in a paralyzed person with anarthria[J]. New England Journal of Medicine, 2021, 385(3): 217-227.
> > > > >
> > > > > [6] Cho C J, Chang E, Anumanchipalli G. Neural latent aligner: cross-trial alignment for learning representations of complex, naturalistic neural data[C]//International Conference on Machine Learning. PMLR, 2023: 5661-5676.
> > > > >
> > > > > [7] Liu R, Azabou M, Dabagia M, et al. Drop, swap, and generate: A self-supervised approach for generating neural activity[J]. Advances in neural information processing systems, 2021, 34: 10587-10599.
> > > > >
> > > > > [8] Wang Y, Li C, Li W, et al. Exploring behavior-relevant and disentangled neural dynamics with generative diffusion models[J]. Advances in Neural Information Processing Systems, 2024, 37: 34712-34736.

---

> > > > > > ### Author Response · Authors · 2025-11-27
> > > > > > **Response to Reviewer xusM by Authors (6/8)**
> > > > > >
> > > > > > **Answer to W5**:
> > > > > >
> > > > > > > ...The discussion of disentanglement representation learning (Section 2.3) mentions several relevant methods (QLAE, Tripod, pi-VAE, PDisVAE) but does not empirically compare against any of them, making it difficult to assess whether DPQ offers advantages over existing disentanglement techniques when applied to neural data...
> > > > > >
> > > > > > Thank you for the helpful suggestion. Since QLAE and Tripod are primarily designed for image datasets, it's difficult to fine-tune their parameters to ensure effective convergence on intracranial neural datasets. Therefore, we mainly consider pi-VAE and PDisVAE as follows.
> > > > > >
> > > > > > We compare DPQ with other methods on the Du-IN 61-word classification task, using the same Neural Encoder as the backbone. Specifically, PDisVAE achieves 64.01%, but we are unable to evaluate pi-VAE due to its requirement of an explicit prior about task structure. pi-VAE and PDisVAE were originally developed for neural spike data. However, **the boundaries of neural events in intracranial LFP recordings (e.g., sEEG, ECoG) are not as clear as those in neural spike recordings**. Therefore, when modeling sEEG/EEG signals [1,2,3], VQ-VAE is adopted to extract discrete neural states by encouraging temporal disentanglement. Besides, unlike VQ-VAE, which provides discrete indices, VAE models provide continuous latents for self-supervision, which offers less effective guidance during MAE training [1,2]. Finally, pi-VAE specifically requires both supervised labels and pre-defined task structures (e.g., 2D motor control, cognitive maps), making it infeasible for speech decoding where words involve complex articulator compositions (e.g., tongue, jaw, lips). **Without explicit priors for speech encoding mechanisms, we cannot infer the necessary task space from words, precluding pi-VAE evaluation.**
> > > > > >
> > > > > > **References**:
> > > > > >
> > > > > > [1] Jiang W B, Zhao L M, Lu B L. Large brain model for learning generic representations with tremendous EEG data in BCI[J]. arXiv preprint arXiv:2405.18765, 2024.
> > > > > >
> > > > > > [2] Zheng H, Wang H T, Jiang W B, et al. Du-IN: Discrete units-guided mask modeling for decoding speech from Intracranial Neural signals[J]. arXiv preprint arXiv:2405.11459, 2024.
> > > > > >
> > > > > > [3] Gui H, Li X, Chen X. Vector quantization pretraining for eeg time series with random projection and phase alignment[C]//International Conference on Machine Learning. PMLR, 2024: 16731-16750.

---

> > > > > > > ### Author Response · Authors · 2025-11-27
> > > > > > > **Response to Reviewer xusM by Authors (7/8)**
> > > > > > >
> > > > > > > **Answer to W6**:
> > > > > > >
> > > > > > > > ...The paper uses spectral clustering with default scikit-learn parameters for grouping channels, but provides no analysis of robustness to this choice—would other clustering algorithms (e.g., hierarchical clustering with different linkage criteria, community detection algorithms, Louvain modularity optimization) yield similar functional groups?
> > > > > > >
> > > > > > > Thank you for the valuable feedback. As demonstrated in **Answer to W1**, our clustering objective is guided by the neuroscientific principle of modular brain organization [1,2], which favors sparse, robust connectivity matrices. Since for different clustering algorithms, the estimated channel connectivity is fixed. Other methods (e.g., hierarchical clustering) yield similar results (Line 214).
> > > > > > >
> > > > > > > > The choice of k=10 clusters appears arbitrary, justified only through post-hoc ablation showing performance is relatively stable across 5-20 clusters, but there is no principled method for selecting k or determining whether the true number of functional groups varies by subject.
> > > > > > >
> > > > > > > Thank you for the helpful suggestion. Actually, **identifying the exact number of clusters is not the core aim of this study**. The core aim of this work is to **provide a practical toolbox for intracranial neural decoding in real-world clinical settings, especially exploratory decoding where labeled data is scarce, and the relevant brain areas may be unknown**, as detailed in **General Rebuttal by Authors**. We provide more results to demonstrate why identifying the exact number of clusters doesn’t affect the core contribution of BrainStratify-Coarse. We demonstrate the Silhouette Coefficient [3] (with the best value as `1` and the worst value as `-1`) of different numbers of clusters based on BrainStratify-Coarse, which is commonly used to select the best number of clusters. Specifically, we normalize the channel attention graph within (0,1) and calculate the channel distance according to $\texttt{dist=1-attn}$, then we evaluate `sklearn.metrics.silhouette_score` on the normalized channel distances.
> > > > > > >
> > > > > > > | # of clusters | 2 | 5 | 10 | 15 | 20 | 25 | 30 | 50 | 100 |
> > > > > > > | --- | --- | --- | --- | --- | --- | --- | --- | --- | --- |
> > > > > > > | Du-IN | 0.22$\pm$0.03 | 0.34$\pm$0.03 | 0.52$\pm$0.04 | 0.63$\pm$0.03 | 0.71$\pm$0.04 | 0.65$\pm$0.04 | 0.50$\pm$0.03 | 0.28$\pm$0.02 | 0.06$\pm$0.01 |
> > > > > > > | Brain Treebank | 0.17$\pm$0.02 | 0.24$\pm$0.03 | 0.38$\pm$0.04 | 0.49$\pm$0.05 | 0.56$\pm$0.04 | 0.60$\pm$0.04 | 0.57$\pm$0.03 | 0.44$\pm$0.03 | 0.11$\pm$0.01 |
> > > > > > >
> > > > > > > Therefore, the best number of clusters is typically within (20,25). This result is expected, as distinct submodules exist within functional groups, e.g., [tongue,jaw,lip]-control submodules within vSMC (for vocal production). Increasing cluster counts beyond `10` yields finer spatial resolution of functional groups while the decoding performance remains similar, as detailed in Figure 10a.
> > > > > > >
> > > > > > > > The weighted anatomical counts metric (Appendix I) is introduced as a quantitative measure of cluster quality, but its interpretation is unclear. why should lower N_anatomy indicate better clustering, and what is the expected value under a null model?
> > > > > > >
> > > > > > > Thank you for the valuable feedback. $N_{a}=1$ indicates perfect anatomical alignment; $N_{a}$ approaching the total region count suggests less successful functional grouping. Thus, lower $N_{a}$ values signify closer correspondence between channel clusters and anatomical labels. The expected value under a null model is 15.25$\pm$1.20 for Du-IN and 19.0$\pm$2.37 for Brain Treebank, which aligns with CCM’s $N_{a}$.
> > > > > > >
> > > > > > > > The paper reports that "most models surpass the PopT baseline with SC selection" when evaluated on channels selected by BrainStratify-Coarse, but this comparison conflates channel selection quality with model performance, and it is unclear whether improvements stem from better channel selection or from the specific channels selected happening to be easier for certain model architectures.
> > > > > > >
> > > > > > > Thank you for the valuable feedback. We compared the performance of PopT under two channel selection strategies to ensure rigorous ablation. The results are reported in Table 3 for Brain Treebank. For Du-IN dataset, we reported the results in Figure 3a. **BrainStratify-Coarse is inspired by modular brain organization [1,2], and we visualize the estimated channel connectivity in Figure 3b to further enhance its explanibility.**
> > > > > > >
> > > > > > > **References**:
> > > > > > >
> > > > > > > [1] Buzsáki G. Rhythms of the Brain[M]. Oxford university press, 2006.
> > > > > > >
> > > > > > > [2] Silva A B, Littlejohn K T, Liu J R, et al. The speech neuroprosthesis[J]. Nature Reviews Neuroscience, 2024, 25(7): 473-492.
> > > > > > >
> > > > > > > [3] Rousseeuw P J. Silhouettes: a graphical aid to the interpretation and validation of cluster analysis[J]. Journal of computational and applied mathematics, 1987, 20: 53-65.

---

> > > > > > > > ### Author Response · Authors · 2025-11-27
> > > > > > > > **Response to Reviewer xusM by Authors (8/8)**
> > > > > > > >
> > > > > > > > **Answer to W7**:
> > > > > > > >
> > > > > > > > > ...While the results on ALS and SCI patients are promising, these are single-subject cases with relatively short-term evaluation, and the paper does not address critical questions for clinical deployment such as long-term stability of the learned functional groups across days or weeks, robustness to electrode impedance changes, or performance degradation in the presence of neural plasticity or disease progression.
> > > > > > > >
> > > > > > > > Thank you for the insightful feedback. In clinical settings, these patients are relatively special and rare, such as ALS patients with long-term implanted electrodes. Most related studies [1-3] have only 1\~2 subjects. Due to the limitations of open-source datasets, subject-specific modeling and evaluation are acceptable based on invasive recordings (e.g., [4] only includes 2 subjects). Besides, previous studies [5,6] typically evaluate neural disentanglement results with real-world neural spike recordings collected from 1~2 NHPs.
> > > > > > > >
> > > > > > > > Furthermore, in the word-reading epidural ECoG dataset, the ECoG was implanted in that subject for approximately **`6` months**. Our evaluation framework adheres to a rigorous clinical simulation by **reserving data exclusively from the final day for testing**. This provides a robust estimate of real-world performance. While long-term implants are uncommon for sEEG, the functional modules we identify are grounded in established neuroscience, which indicates that such neural organizations are fundamentally stable over time.
> > > > > > > >
> > > > > > > > > The claim that epidural ECoG is "minimally invasive" and "reduces tissue damage"...adaptation effects.
> > > > > > > >
> > > > > > > > Thank you for the valuable feedback. As detailed in **General Rebuttal by Authors**, BrainStratify addresses a pressing need: it provides a practical toolbox for intracranial neural decoding in real-world clinical settings, especially exploratory decoding where labeled data is scarce, and the relevant brain areas may be unknown. By enabling neuroscience researchers to fully exploit their valuable and hard-won datasets, our framework will potentially accelerate the discovery process in cognitive state decoding. For example, in some early speech decoding studies [7], researchers were not aware of the importance of channel selection based on functional similarity, resulting in not fully exploiting the potential of the collected data.
> > > > > > > >
> > > > > > > > **References**:
> > > > > > > >
> > > > > > > > [1] Metzger S L, Littlejohn K T, Silva A B, et al. A high-performance neuroprosthesis for speech decoding and avatar control[J]. Nature, 2023, 620(7976): 1037-1046.
> > > > > > > >
> > > > > > > > [2] Willett F R, Kunz E M, Fan C, et al. A high-performance speech neuroprosthesis[J]. Nature, 2023, 620(7976): 1031-1036.
> > > > > > > >
> > > > > > > > [3] Moses D A, Metzger S L, Liu J R, et al. Neuroprosthesis for decoding speech in a paralyzed person with anarthria[J]. New England Journal of Medicine, 2021, 385(3): 217-227.
> > > > > > > >
> > > > > > > > [4] Cho C J, Chang E, Anumanchipalli G. Neural latent aligner: cross-trial alignment for learning representations of complex, naturalistic neural data[C]//International Conference on Machine Learning. PMLR, 2023: 5661-5676.
> > > > > > > >
> > > > > > > > [5] Liu R, Azabou M, Dabagia M, et al. Drop, swap, and generate: A self-supervised approach for generating neural activity[J]. Advances in neural information processing systems, 2021, 34: 10587-10599.
> > > > > > > >
> > > > > > > > [6] Wang Y, Li C, Li W, et al. Exploring behavior-relevant and disentangled neural dynamics with generative diffusion models[J]. Advances in Neural Information Processing Systems, 2024, 37: 34712-34736.
> > > > > > > >
> > > > > > > > [7] Angrick M, Ottenhoff M C, Diener L, et al. Real-time synthesis of imagined speech processes from minimally invasive recordings of neural activity[J]. Communications biology, 2021, 4(1): 1055.

---

### Official Review · Reviewer_721d · 2025-10-30

**Soundness:** 2
**Presentation:** 2
**Contribution:** 3
**Rating:** 2
**Confidence:** 4

**Summary:**

The paper presents a unified framework for speech decoding via disentangling fine and coarse neural information from intracranial recordings in two complementary stages, i.e., focusing on spatial and temporal information, respectively. Its effectiveness is evaluated comprehensively across six datasets consisting of different signal types and tasks

**Strengths:**

The proposed method for neural disentanglement demonstrates improved performance over previous methods considered. Also the framework is generalizable to different tasks beyond speech decoding and various types of biosignals. Model configuration is well written in detail and also the authors claim the dataset and code will be publicly available upon publication.

**Weaknesses:**

The framing of ‘speech decoding’ as presented is confusing, and can be a bit misleading based on the downstream tasks actually performed. It looks like ‘speech decoding’ is just one part of downstream tasks, and the proposed framework looks like it is suitable for other types of tasks, too, so I do not see any part of the framework that is specific to speech.

Also, from results (Table 2 and 3), it is difficult to capture the effectiveness of ‘BrainStra.-Coarse’. It appears that the only rows to compare the ‘BrainStra.-Coarse’ condition are the last two rows of Table 2, but it presents contradicting results. A review of additional results in the Appendix did not appear to show any firm evidence to support the effectiveness of ‘Coarse’.

Regarding the articulatory regions (tongue, lips, and jaw), it is not clear how those data are annotated. According to the description, it looks like subjects were instructed to ‘intend’ to move those parts, but unclear whether there was actual articulatory movements. Also, neural underpinning and muscles involved in movements of different parts of tongue (e.g., tongue tip, body, or root) are slightly different, but this is either not properly addressed or if only the tongue tip is taken for analysis. If it is the latter case, I believe it should be clearly stated that ‘tongue tip’, not just ‘tongue’ in general. Also, those articulatory movements are correlated to each other, that is, for instance, jaw or lip movements are related to each other, rather than totally separate, but these points were not properly addressed in the paper.

Some results in some tables do not indicate which types of metrics were used for evaluation. It looks accuracy, but they may also be balanced accuracy, F1 scores, or some other metric,  what exact metrics are being reported is not clear.

While the topic is interesting and important, I think this paper is not yet ready for publication and would benefit from significant revision.

**Questions:**

Were there any methods to address subject invariability or generalizability? Also, were there any analysis on different performance among the subjects?
Why is this framework framed as ‘speech decoding’ as it is not speech-specific and generalizable to other tasks too?
What do the authors intend by a ‘unified’ framework?

**Details Of Ethics Concerns:**

I do not see any major ethics concerns, but as it involves human subjects, an additional ethics review by designated ethics reviewers would be beneficial.

---

> ### Author Response · Authors · 2025-11-15
> **Response to Reviewer 721d by Authors (1/2)**
>
> Thank you for the thorough review and constructive comments. We are deeply grateful for acknowledging our method "is generalizable to different tasks beyond speech decoding and various types of biosignals". Please see below for the point-by-point responses to your comments.
>
> **Answer to W1**:
>
> Thank you for the insightful feedback. We want to clarify the rationale for limiting our claims to speech decoding.
>
> First, it’s worth noting that, due to the lack of publicly available intracranial datasets, **our evaluation primarily focuses on speech decoding tasks (e.g., vocal production and speech perception)**, which constitute 4 of our 6 benchmark tasks.
>
> Second, the core design of BrainStratify-Coarse is inspired by the brain's modular organization. **This design principle is largely consistent with research on the neural coding of cognitive function organization, which aims to break down cognitive function and identify the stable and precise information encoded by each brain region (such as vSMC encoding speech motor and STG encoding auditory perception).** For such cognitive-decoding tasks, our method aims to estimate the precise functional boundaries of these modules, as demonstrated in Figure 3, thus avoiding the involvement of irrelevant channels to reduce performance (Du-IN’s Figure 4). **However, decoding tasks are not limited to cognitive functions; there are other types of decoding tasks, such as epilepsy detection and sleep staging. For these types of tasks, SOTA decoding models are primarily based on the neurofoundation models [1,2].** Therefore, our model is not capable of handling all types of decoding tasks, but focuses on decoding tasks related to cognitive functions (e.g., speech decoding). Besides, our method can potentially help exploratory neural cognitive-decoding tasks, whose involved brain regions are unclear, as discussed in “General Rebuttal by Authors”.
>
> Third, to better understand the disentanglement of BrainStratify-Fine, we need to collect datasets with the simplified task space (Section 4.5) according to previous neural encoding studies [3]. **Due to the lack of more public datasets suitable for evaluating disentanglement, we presently confine our claim to the speech decoding field, maintaining a rigorous and focused claim.** That said, we agree that generalization to other tasks is key. We preliminarily evaluated our framework on an ECoG motor-intention task and an EMG-based typing task. The positive decoding (not disentanglement) results on these non-language tasks suggest promising potential for broader extrapolation, and non-language tasks are worth systematically exploring if the corresponding datasets for studying encoding mechanisms are available.
>
> **Answer to W2**:
>
> Thank you for the valuable feedback. As elaborated in our "General Rebuttal by Authors", this work presents a comprehensive toolbox designed to address the key challenges of real-world clinical settings, where the vast majority of intracranial recordings are unlabeled and task-specific data is scarce. To further substantiate this, we have included additional results (extending Figure 1) that demonstrate how BrainStratify-Coarse **significantly reduces the number of supervised samples required to select task-relevant channels, achieving the data-efficient channel selection for sEEG in clinical settings** (Line 55 & Line 378).
>
> | # of samples | 600 | 1200 | 1800 | 2400 | 3000 |
> | --- | --- | --- | --- | --- | --- |
> | BrainStratify-Coarse | 58.02$\pm$4.27 | 58.02$\pm$4.27 | 58.02$\pm$4.27 | 58.02$\pm$4.27 | 58.02$\pm$4.27 |
> | MC | 15.92$\pm$3.18 | 24.45$\pm$3.67 | 45.84$\pm$6.26 | 53.24$\pm$4.78 | 58.06$\pm$4.19 |
>
> **Answer to W3**:
>
> Thank you for the insightful feedback. We strictly adhered to the experimental setup of [3], where participants performed attempted (non-vocalized) movements. Regarding the tongue, our evaluation indeed focused on the front tongue as defined in [3], and we will revise this description for clarity.
>
> Crucially, **as explored in [3], the spatial resolution of ECoG data seems insufficient to discriminate neural encoding between the front and back tongue**. Therefore, our analysis focused primarily on the codebook's encoding of the broader articulators: the jaw, tongue, and lips.
>
> While we acknowledge potential co-activation between the jaw and lips, the underlying muscle control is distinct. For instance, jaw movement typically occurs with relative relaxation of the lip muscles. **This functional distinction is supported by the clearly separable neural encoding patterns for these articulators demonstrated in Figure 1d of [3].**

---

> ### Author Response · Authors · 2025-11-15
> **Response to Reviewer 721d by Authors (2/2)**
>
> **Answer to W4**:
>
> Thank you for the helpful feedback. We strictly follow the standard evaluation protocols for the Du-IN and Brain Treebank datasets. For classification tasks (e.g., word classification in Du-IN), we report accuracy. For binary classification tasks (e.g., state classification in the Brain Treebank), we report ROC-AUC. For sequential classification tasks (e.g., syllable sequential classification in Du-IN), we report $\texttt{1-SER}$, where $\texttt{SER}$ means syllable error rate. For regression tasks (e.g., speech synthesis), we report the Pearson Correlation Coefficient. Please see Appendix B for more details. **We revised the table captions to avoid any misunderstanding (Table 2).**
>
> **Answer to Q1**:
>
> Thank you for the valuable feedback. **Some of the baselines evaluated in this work (e.g., PopT, H2DiLR) aim to tackle the issue related to subject invariability or generalizability.** Specifically, PopT pre-trains a neurofoundaiton model across multiple subjects via self-supervision, learning general representations for downstream subject-specific fine-tuning. H2DiLR pre-trains a temporal model to capture the shared brain dynamics via self-supervision, thus improving subject-specific fine-tuning. We either directly download official code from GitHub or acquire the code from the corresponding authors to evaluate them. As reported in Tables 2 & 3, our method achieves the best performance, which is reproducible as we provide the model and checkpoints in the anonymous GitHub link.
>
> **Answer to Q2**:
>
> Thank you for the valuable feedback. As reported in Du-IN, Brain Treebank, etc., the performance difference among subjects mainly stems from the significant differences in implantation sites among the subjects, as the electrode implantation site is determined by the location of the epilepsy. This is why **research in this field often requires channel selection to remove irrelevant channels, thus improving decoding performance**. This is also the problem this work aims to solve: **by leveraging the modular brain computation, our method significantly reduces the amount of supervised samples required for channel selection, which is crucial in clinical settings**. Please see “General Rebuttal by Authors” for more details.
>
> **Answer to Q3**:
>
> Thank you for the insightful feedback. Please see “Answer to W1” and “General Rebuttal by Authors” for more details.
>
> **Answer to Q4**:
>
> Thank you for the insightful feedback. Since speech decoding tasks (e.g., vocal production, speech perception) are cognitive-decoding tasks, these decoding tasks can all be broken down into two independent problems: (1) **channel selection**, used to determine the boundaries of functional groups; and (2) **general representation learning**, used to understand the structure of neural dynamics within functional groups, thus enhancing diverse decoding paradigms (e.g., classification, sequential classification, regression). The term “unified” means we aim to create a unified framework that systematically solves these two problems to achieve optimal decoding performance. Please see “General Rebuttal by Authors” for more details.
>
> **References**:
>
> [1] Yuan Z, Shen F, Li M, et al. Brainwave: A brain signal foundation model for clinical applications[J]. arXiv preprint arXiv:2402.10251, 2024.
>
> [2] Jiang W B, Zhao L M, Lu B L. Large brain model for learning generic representations with tremendous EEG data in BCI[J]. arXiv preprint arXiv:2405.18765, 2024.
>
> [3] Metzger S L, Littlejohn K T, Silva A B, et al. A high-performance neuroprosthesis for speech decoding and avatar control[J]. Nature, 2023, 620(7976): 1037-1046.

---

### Official Review · Reviewer_cnbL · 2025-10-31

**Soundness:** 3
**Presentation:** 3
**Contribution:** 2
**Rating:** 4
**Confidence:** 3

**Summary:**

The authors propose a two-stage intracranial speech decoding method that (1) identifies relevant channels in the recording via a neural clustering approach and (2) applies a regular deep learning method to classify from the selected channels. Stage (1) does not seem to help improve downstream performance, though stage (2) shows interesting relationships between the learned codex and articulatory actions and performs well in comparison to alternative approaches.

**Strengths:**

- Identifying relevant channels is interesting and a potentially useful procedure for improving downstream performance
- Interesting analyses of learned codex signals directly relating to articulatory actions
- Good use of statistical significance and error bars in all analyses

**Weaknesses:**

- Improvement over multi-channel with the channel selection method is weak based on ablations, suggesting it may not be helpful for downstream decoding (though it does not hurt)
- Since BrainStratify-Fine is essentially a supervised method, could the authors please also compare to a simple supervised deep learning baseline, e.g. training EEGNet and an MLP baseline
- Minor: Line 387. Suppresses → surpasses
- Table 2 and 3 colouring is misleading. Green should not indicate lack of statistical significance

I am willing to raise my score if the authors can address the weaknesses satisfactorily.

**Questions:**

- When evaluating the pre-trained models, did the authors consider end-to-end fine-tuning them rather than linear probing? This might be a fairer comparison to the end-to-end training used for BrainStratify-Fine

---

> ### Author Response · Authors · 2025-11-15
> **Response to Reviewer cnbL by Authors**
>
> Thank you for the thorough review and constructive comments. We are deeply grateful for acknowledging that "identifying relevant channels is interesting and a potentially useful procedure for improving downstream performance" and “interesting analyses of learned codex signals directly relating to articulatory actions". Please see below for the point-by-point responses to your comments.
>
> **Answer to W1**:
>
> Thank you for the valuable feedback. As elaborated in our "General Rebuttal by Authors", this work presents a comprehensive toolbox designed to address the key challenges of real-world clinical settings, where the vast majority of intracranial recordings are unlabeled and task-specific data is scarce. To further substantiate this, we have included additional results (extending Figure 1) that demonstrate how BrainStratify-Coarse **significantly reduces the number of supervised samples required to select task-relevant channels, achieving the data-efficient channel selection for sEEG in clinical settings**.
>
> | # of samples | 600 | 1200 | 1800 | 2400 | 3000 |
> | --- | --- | --- | --- | --- | --- |
> | BrainStratify-Coarse | 58.02$\pm$4.27 | 58.02$\pm$4.27 | 58.02$\pm$4.27 | 58.02$\pm$4.27 | 58.02$\pm$4.27 |
> | MC | 15.92$\pm$3.18 | 24.45$\pm$3.67 | 45.84$\pm$6.26 | 53.24$\pm$4.78 | 58.06$\pm$4.19 |
>
> **Answer to W2**:
>
> Thank you for the valuable feedback. The performance on the Du-IN dataset is 22.8% (EEGNet) and 10.2% (MLP), respectively. The EEGNet performance aligns with that of TS-TCC (as detailed in Du-IN's Table 2), which also uses a CNN backbone. The extremely low performance of MLP is expected. Since the neural activity of vocal production is not well-aligned with the onset of stimulus. Decoding speech typically requires modeling temporal relationships to achieve better performance.
>
> **Answer to W3**:
>
> Thank you for the valuable feedback. We fixed this typo in the updated version.
>
> **Answer to W4**:
>
> Thank you for the helpful suggestion. We changed it to gray for better illustration in the updated version.
>
> **Answer to Q1**:
>
> Thank you for the valuable feedback. We strictly follow the **end-to-end fine-tuning pipeline** of neurofoundation models (e.g., PopT [1], LaBraM [2]) to ensure fair comparison. BrainStratify outperformed all foundation models on the widely-used cognitive state sEEG datasets (i.e., Du-IN for vocal production and Brain Treebank for speech perception) under fair comparison, and the core contribution of this work lies in how to best integrate the strengths of these two types of models by constructing a general framework to achieve optimal performance in speech decoding and other similar tasks (e.g., motor intention) in clinical settings.
>
> Besides, we further provide linear probing results on Brain Treebank dataset, for comparison with PopT’s linear probing results (PopT’s Figure 17). All checkpoints are available on the anonymous GitHub repository.
> | Pitch | Volume | Sent. Onset | Word Onset |
> | --- | --- | --- | --- |
> | 0.74$\pm$0.01 | 0.84$\pm$0.02 | 0.91$\pm$0.02 | 0.92$\pm$0.01 |
>
> In the Brain Treebank dataset, our frozen model achieves great improvement (e.g., 0.91v.s.0.75 on sentence onset detection) compared to the linear probing results reported in PopT Figure 17. Besides, **the parallel work [6] from the PopT team also questions the effectiveness of representation learning on neurofoundation models via linear probing**.
>
> **References**:
>
> [1] Chau G, Wang C, Talukder S, et al. Population transformer: Learning population-level representations of neural activity[J]. ArXiv, 2025: arXiv: 2406.03044 v4.
>
> [2] Jiang W B, Zhao L M, Lu B L. Large brain model for learning generic representations with tremendous EEG data in BCI[J]. arXiv preprint arXiv:2405.18765, 2024.
>
> [3] Zheng H, Wang H, Jiang W, et al. Du-IN: Discrete units-guided mask modeling for decoding speech from Intracranial Neural signals[J]. Advances in Neural Information Processing Systems, 2024, 37: 79996-80033.
>
> [4] Zahorodnii A, Wang C, Stankovits B, et al. Neuroprobe: Evaluating Intracranial Brain Responses to Naturalistic Stimuli[J]. arXiv preprint arXiv:2509.21671, 2025.
>
> [5] Zhang D, Yuan Z, Yang Y, et al. Brant: Foundation model for intracranial neural signal[J]. Advances in Neural Information Processing Systems, 2023, 36: 26304-26321.
>
> [6] Zahorodnii A, Wang C, Stankovits B, et al. Neuroprobe: Evaluating Intracranial Brain Responses to Naturalistic Stimuli[J]. arXiv preprint arXiv:2509.21671, 2025.

---

> > ### Comment · Reviewer_cnbL · 2025-11-17
> > **Response to Author Rebuttal**
> >
> > Thank you for your response. You have addressed my questions and main concerns. I have raised my score accordingly.

---

> > > ### Author Response · Authors · 2025-11-17
> > > **Official Comment by Authors**
> > >
> > > We sincerely thank the reviewer again for all their constructive comments and are grateful for the positive assessment of this work!

---

### Official Review · Reviewer_5Gy2 · 2025-11-07

**Soundness:** 2
**Presentation:** 3
**Contribution:** 2
**Rating:** 4
**Confidence:** 4

**Summary:**

The paper proposes BrainStratify, a two-stage framework for speech decoding from intracranial signals (sEEG/ECoG): Coarse stage learns an inter-channel attention graph with a spatial-context objective and clusters channels into functional groups; Fine stage introduces Decoupled Product Quantization (DPQ) within a VQ-VAE/MAE pipeline to disentangle neural components and guide masked code prediction.

**Strengths:**

1. The paper proposes a clear, modular design that mirrors neurophysiology intuition (group → sub-component). The pipeline is well explained with task definitions and training details.

2. The paper is well written. I particularly liked the codex-wise analyses on the articulation benchmark (jaw/lips/tongue) that add interpretability and are a nice touch toward making the discrete units meaningful.

**Weaknesses:**

1. Novelty is incremental relative to known building blocks. There is little algorithmic innovation beyond integrating well-known components; DPQ is essentially PQ with independence encouragement and used to supervise MAE via code indices.

2. The new epidural ECoG datasets each contain one subject; most modeling is subject-specific. Therefore, the claims of robust generalization and neuroscience-inspired modularity are not convincingly demonstrated across patients or sessions. Cross-subject transfer (train on N−1, test on held-out subject) would better substantiate robustness.

3. While conceptually appealing, this two-stage design is highly complex, integrating transformers, convolutional encoders, spectral clustering, a VQ-VAE with multiple codebooks, and a masked autoencoding step. The sheer number of components makes it difficult to pinpoint which aspects drive improvements.

4. Moreover, certain design choices are not fully justified. The authors mention adding “either learnable or MNI-based” spatial embeddings before the spatial transformer, implying a choice between using standardized brain coordinates (MNI space) or learning positional encodings. It remains unclear which option was used in final experiments and how much this choice impacts performance or interpretability.

Without explicit ablations or rationale for these architectural decisions, the rigor of the design is in question. The method appears over-engineered, stitching together many recent techniques (transformers, product quantization, masked modeling) and this raises the possibility that a far simpler architecture might achieve similar results.

Minor comment: Fig. 2 caption is missing (c).

**Questions:**

I am a bit curious as to how the functional group of channels is ultimately selected for decoding. The coarse stage clusters electrodes into groups, but the paper admits that it select[s] and combine[s] these groups "based on their performance in downstream decoding tasks.”. In other words, after unsupervised clustering, the authors use supervised task performance to decide which cluster (or combination of clusters) is relevant. This procedure risks information leakage or overfitting if not handled carefully. It blurs the line between an unsupervised pretext step and the supervised evaluation. The paper does not detail whether this selection was done using a held-out validation set, how many labeled examples were used, or how they avoided biasing the final results. Can the authors please explain this?

---

> ### Author Response · Authors · 2025-11-15
> **Response to Reviewer 5Gy2 by Authors (1/3)**
>
> Thank you for the thorough review and constructive comments. We are deeply grateful for acknowledging our method "proposes a clear, modular design that mirrors neurophysiology intuition (group $\rightarrow$ sub-component)" and "the codex-wise analyses add interpretability and are a nice touch toward making the discrete units meaningful". Please see below for the point-by-point responses to your comments.
>
> **Answer to W1**:
>
> Thank you for the valuable feedback. Please see “General Rebuttal by Authors” for a detailed explanation of the novelty of this work. The aim of this work is to build a unified framework for intracranial neural decoding, especially for speech decoding.
>
> As for the disentanglement representation learning part, previous studies [1-3] on intracranial neural decoding rarely considered the issue of neural component entanglement. They primarily focused on either directly performing general pre-training [1,2] or attempting to learn shared neural dynamics among subjects [3]. The core contribution of our work lies in **exploring this issue in the field of intracranial neural decoding** and achieving SOTA decoding performance on widely used intracranial neural decoding datasets (Du-IN [1] and Brain Treebank [4]) using DPQ. To systematically enable this line of inquiry, we **collected a articulation control dataset (Section 4.5) as a benchmark** and validated the feasibility of our method through detailed codex-wise analysis.
>
> **Answer to W2**:
>
> Thank you for the valuable feedback. In clinical settings, these patients are relatively special and rare, such as ALS patients with long-term implanted electrodes. Most related studies [5-7] have only 1\~2 subjects. Due to the limitations of open-source datasets, subject-specific modeling and evaluation are acceptable based on invasive recordings (e.g., [8] only includes 2 subjects). Besides, previous studies [9,10] typically evaluate neural disentanglement results with real-world neural spike recordings collected from 1~2 NHPs. We additionally provide the disentanglement results on our collected motor intention dataset as follows to further demonstrate the effectiveness. Specifically, we evaluate the three 2-way classification tasks of right-[hand,elbow,leg], with left-finger as the other state, and report the absolute accuracy improvement against the chance level (50%) for better illustration. The codex-wise contributions of right-[hand,elbow] and right-leg vary, demonstrating that DPQ can distinguish motor intentions with large differences.
>
> | Codex | 1 | 2 | 3 | 4 | 5 | 6 | 7 | 8 |
> | --- | --- | --- | --- | --- | --- | --- | --- | --- |
> | Right-Hand | 4.55$\pm$1.22 | **22.71$\pm$2.91** | 6.94$\pm$2.19 | **18.72$\pm$1.59** | 8.79$\pm$1.68 | 5.19$\pm$2.01 | 6.08$\pm$1.12 | **14.19$\pm$1.29** |
> | Right-Elbow | 2.10$\pm$0.79 | **20.09$\pm$1.75** | 7.16$\pm$1.22 | **15.92$\pm$0.94** | 6.73$\pm$1.44 | 6.11$\pm$0.92 | 5.70$\pm$1.32 | **17.52$\pm$2.19** |
> | Right-Leg | 1.29$\pm$0.52 | **25.91$\pm$1.68** | **20.16$\pm$1.66** | 8.54$\pm$2.41 | 3.16$\pm$1.19 | 9.17$\pm$2.04 | 7.26$\pm$1.41 | 5.70$\pm$1.81 |
>
> As demonstrated in **General Rebuttal by Authors**, this work aims to tackle two fundamental challenges in intracranial BCIs: (1) data-efficient channel selection for sEEG (Line 55), and (2) learning generalizable representations that enhance performance across diverse decoding paradigms (e.g., classification, regression, sequential classification) in intracranial recordings (e.g., sEEG, ECoG). As our model is subject-specific, **the claimed generalization across different decoding paradigms (Tables 2 & 12) and neuroscience-inspired modularity (Figure 3) are evaluated within the subject**. Please see Appendix N & O for more details on the subject-wise evaluation. Moreover, cross-subject transfer -- training one neurofoundation model to support general channel clustering and neural decoding -- is not the core aim of this study. Previous studies have attempted to build intracranial neurofoundation models for general neural decoding, and we provide a comprehensive comparison in Tables 2 & 3. This work aims to address real-world clinical settings, as detailed in “General Rebuttal by Authors”.

---

> ### Author Response · Authors · 2025-11-15
> **Response to Reviewer 5Gy2 by Authors (2/3)**
>
> **Answer to W3**:
>
> Thank you for the valuable feedback. The core difference of our two-stage framework over from multi-step pre-trained models [11] is that the two steps are performed independently, with each designed to address a different problem in intracranial neural decoding. This allows us to independently evaluate the importance of the components within each stage. Please refer to Appendix K for detailed ablations.
>
> BrainStratify-Coarse leverages well-established components (CNN, Transformer) within a novel, purpose-built framework. Please see Section 3.2 for the detailed design. Its core innovation lies in the design of a spatial context discrimination task that converges efficiently on limited sEEG data. This enables us to derive a channel connectivity graph from spatial attention scores, which we then use to validate the brain's modular organization through clustering and visualization. For BrainStratify-Fine, we designed DPQ based on VQ-VAE to operate effectively within these identified functional groups.
>
> As for concerns related to whether a far simpler architecture might achieve similar results, please see **General Rebuttal by Authors** for more details. For data-efficient channel selection, we are the first to extract and validate such functional groups directly from model attention, please see Figure 1 & 3 for details. For neural decoding, we demonstrate advanced performance compared to previous baselines in Tables 2 & 3.
>
> **Answer to W4**:
>
> Thank you for the valuable feedback. To clarify, we conducted experiments using both approaches and observed no difference in the clustering outcomes, which aligns with findings on neural decoding from prior work [12]. As such, we reported the results based on the learnable positional encoding for consistency. The complete implementation is available in the provided anonymous GitHub for verification.
>
> **Answer to W5**:
>
> Thank you for the valuable feedback. Please refer to “Answer to W3” and “General Rebuttal by Authors”.
>
> **Answer to W6**:
>
> Thank you for the helpful suggestion. We have revised the corresponding figure captions: (c) Overview of downstream neural decoding tasks.
>
> **Answer to Q1**:
>
> Thank you for the helpful suggestion. BrainStratify-Coarse aims to identify the fine-grained boundary of functional groups based on intracranial sEEG recordings (with high spatial resolution), which provide better clustering labels for downstream channel selection compared to anatomical region labels. After clustering channels into functional groups based on the inter-channel attention graph learned via pure self-supervision, the channel groups are fixed. We directly train and evaluate neural decoding model with 600 samples (**Figure 1 & Line 378**) to rank channel clusters. Here, we demonstrate the ranked cluster-wise performance (w/ 600 samples) on Du-IN and Brain Treebank (sentence onset detection) datasets. We can see that even with limited label data, we are still able to determine which clusters the task information is primarily encoded in due to the modular brain computation.
>
> | Cluster Index | 1 | 2 | 3 | 4 | 5 | 6 | 7 | 8 | 9 | 10 |
> | --- | --- | --- | --- | --- | --- | --- | --- | --- | --- | --- |
> | Du-IN (Accuracy (%)) | 24.86$\pm$2.19 | 6.47$\pm$1.02 | 3.52$\pm$0.34 | 2.55$\pm$0.21 | 2.19$\pm$0.22 | 2.02$\pm$0.19 | 1.97$\pm$0.15 | 1.84$\pm$0.15 | 1.82$\pm$0.14 | 1.79$\pm$0.15 |
>
> | Cluster Index | 1 | 2 | 3 | 4 | 5 | 6 | 7 | 8 | 9 | 10 |
> | --- | --- | --- | --- | --- | --- | --- | --- | --- | --- | --- |
> | Brain Treebank (ROC-AUC) | 0.86$\pm$0.03 | 0.67$\pm$0.02 | 0.62$\pm$0.02 | 0.61$\pm$0.02 | 0.58$\pm$0.01 | 0.57$\pm$0.02 | 0.56$\pm$0.01 | 0.56$\pm$0.02 | 0.55$\pm$0.01 | 0.54$\pm$0.01 |
>
> Besides, we additionally perform K-fold evaluation (K=10) on both Du-IN and Brain Treebank datasets. Specifically, we randomly split the dataset of each subject into 10 parts, then combine 8 parts as the training set, leave 2 parts for evaluation and testing, respectively. Then, we **sampled 600 samples from the training set for channel selection, thus avoiding potential information leakage**. Based on the great performance difference among channel groups, this evaluation strategy results in the same selected channels and downstream performance.

---

> > ### Author Response · Authors · 2025-11-29
> > **Response to Reviewer 5Gy2 by Authors (3/3)**
> >
> > **References**:
> >
> > [1] Zheng H, Wang H, Jiang W, et al. Du-IN: Discrete units-guided mask modeling for decoding speech from Intracranial Neural signals[J]. Advances in Neural Information Processing Systems, 2024, 37: 79996-80033.
> >
> > [2] Chau G, Wang C, Talukder S, et al. Population transformer: Learning population-level representations of neural activity[J]. ArXiv, 2025: arXiv: 2406.03044 v4.
> >
> > [3] Wu D, Li S, Feng C, et al. Towards Homogeneous Lexical Tone Decoding from Heterogeneous Intracranial Recordings[J]. arXiv preprint arXiv:2410.12866, 2024.
> >
> > [4] Wang C, Yaari A, Singh A, et al. Brain treebank: Large-scale intracranial recordings from naturalistic language stimuli[J]. Advances in Neural Information Processing Systems, 2024, 37: 96505-96540.
> >
> > [5] Metzger S L, Littlejohn K T, Silva A B, et al. A high-performance neuroprosthesis for speech decoding and avatar control[J]. Nature, 2023, 620(7976): 1037-1046.
> >
> > [6] Willett F R, Kunz E M, Fan C, et al. A high-performance speech neuroprosthesis[J]. Nature, 2023, 620(7976): 1031-1036.
> >
> > [7] Moses D A, Metzger S L, Liu J R, et al. Neuroprosthesis for decoding speech in a paralyzed person with anarthria[J]. New England Journal of Medicine, 2021, 385(3): 217-227.
> >
> > [8] Cho C J, Chang E, Anumanchipalli G. Neural latent aligner: cross-trial alignment for learning representations of complex, naturalistic neural data[C]//International Conference on Machine Learning. PMLR, 2023: 5661-5676.
> >
> > [9] Liu R, Azabou M, Dabagia M, et al. Drop, swap, and generate: A self-supervised approach for generating neural activity[J]. Advances in neural information processing systems, 2021, 34: 10587-10599.
> >
> > [10] Wang Y, Li C, Li W, et al. Exploring behavior-relevant and disentangled neural dynamics with generative diffusion models[J]. Advances in Neural Information Processing Systems, 2024, 37: 34712-34736.
> >
> > [11] Dong Z, Li R, Chong J S X, et al. Brain harmony: A multimodal foundation model unifying morphology and function into 1D tokens[J]. Advances in Neural Information Processing Systems, 2025.
> >
> > [12] Mentzelopoulos G, Chatzipantazis E, Ramayya A G, et al. Neural decoding from stereotactic EEG: accounting for electrode variability across subjects[J]. Advances in Neural Information Processing Systems, 2024, 37: 108600-108624.

---

### Author Response · Authors · 2025-11-15
**General Rebuttal by Authors (1/2)**

Thanks to all reviewers for careful reading and thoughtful feedback. We are grateful for acknowledging our method presents “a clear, modular design that mirrors neurophysiology intuition (group $\rightarrow$ sub-component)”,“the codex-wise analyses on the articulation benchmark (jaw/lips/tongue) that add interpretability and are a nice touch toward making the discrete units meaningful” and “identifying relevant channels is interesting and a potentially useful procedure for improving downstream performance” (Reviewer 5Gy2 & Reviewer cnbL). We are also excited to hear that our work “is generalizable to different tasks beyond speech decoding and various types of biosignals” and “the topic is interesting and important” (Reviewer 721d). Further, we are delighted to hear our method “mirrors hierarchical brain organization principles” (Reviewer xusM).

In this rebuttal, we will clarify the problems our work aims to solve, underscore their importance, and further detail our method's novelty in this context.

# The problems and importance

This work aims to tackle two fundamental challenges in intracranial BCIs: (1) **data-efficient channel selection for sEEG** (Line 55), and (2) **learning generalizable representations that enhance performance across diverse decoding paradigms** (e.g., classification, regression, sequential classification) in intracranial recordings (e.g., sEEG, ECoG).

The brain's modular computation means task-relevant signals are typically distributed sparsely across the brain [1-4]. Since sEEG recordings capture highly localized and non-redundant signals, selecting task-relevant channels is critical for both decoding performance and computational efficiency, especially in complex tasks like word classification in [1]’s Figure 4b. However, current methods (e.g., SC and MC) rely heavily on supervised data, which is a major limitation in clinical practice where the vast majority of intracranial recordings are unlabeled.

This reality exposes a key gap: few works have attempted to systematically address channel selection in a realistic, label-scarce clinical setting. Our work directly confronts this by asking:

1. Can we leverage abundant unlabeled data to drastically reduce the labeled sample requirement for channel selection?

2. Can we learn a unified representation from unlabeled data that boosts performance across diverse decoding paradigms (e.g., classification, sequence labeling, regression)?

For vocal production in the Du-IN dataset [1], different decoding paradigms mean word classification, syllable sequential classification (Table 2), and speech synthesis (Table 12). By solving these problems, we enable more efficient and high-performance decoders, which is a crucial step toward practical clinical BCIs.

# The novelty of our method

Current research on intracranial decoding (especially speech decoding) primarily follows two paths: spatiotemporal models to build neurofoundation models [2,3] and temporal models to model brain dynamics [1,4]. While temporal models excel in cognitive-state decoding performance, they are notoriously labeled-data-hungry for channel selection. Spatiotemporal models, though less effective for intracranial cognitive-state decoding, possess a unique strength: an inherent flexibility in modeling channel dependencies. We leverage this specific strength to validate our novel functional-group-based channel selection approach.

Our work, BrainStratify, integrates these approaches to create a unified framework, which combines the temporal modeling strengths of temporal models with the powerful spatial dependency capture of spatiotemporal models, achieving optimal performance in speech decoding and other similar tasks (e.g., motor intention).

One of our key innovations lies in using the spatial attention graphs from an unsupervised spatiotemporal Transformer to identify functional channel groups that align with the brain's modular organization (Figure 3). To our knowledge, this is the first work to **extract and validate such functional groups directly from model attention, a method that proves highly data-efficient**. This approach drastically reduces the labeled data required for channel selection -- e.g., **from 3,000 to 600 samples in the Du-IN dataset [1] (Figure 1 & Line 378)**. This is **of immediate practical value, as many clinical datasets are similarly label-scarce** (e.g., ~1,500 samples in H2DiLR [4], ~500 samples in Singh et al. [7], ~500 samples in Angrick et al. [5]).

Furthermore, while disentangled representation learning is established in computer vision and emerging in neural encoding studies with neural spike data [6], we are the first to adapt it for intracranial decoding in a clinical BCI context. We specifically **collected an articulation benchmark to quantitatively validate its interpretability via codex-wise analysis**.

---

> ### Author Response · Authors · 2025-11-15
> **General Rebuttal by Authors (2/2)**
>
> In summary, BrainStratify addresses a pressing need: it provides **a practical toolbox for intracranial neural decoding in real-world clinical settings, especially exploratory decoding where labeled data is scarce and the relevant brain areas may be unknown**. By enabling neuroscience researchers to fully exploit their valuable and hard-won datasets, our framework will potentially accelerate the discovery process in cognitive state decoding. For example, in some early speech decoding studies [5], researchers were not aware of the importance of channel selection based on functional similarity, resulting in not fully exploiting the potential of the collected data.
>
> **References**:
>
> [1] Zheng H, Wang H, Jiang W, et al. Du-IN: Discrete units-guided mask modeling for decoding speech from Intracranial Neural signals[J]. Advances in Neural Information Processing Systems, 2024, 37: 79996-80033.
>
> [2] Chau G, Wang C, Talukder S, et al. Population transformer: Learning population-level representations of neural activity[J]. ArXiv, 2025: arXiv: 2406.03044 v4.
>
> [3] Mentzelopoulos G, Chatzipantazis E, Ramayya A G, et al. Neural decoding from stereotactic EEG: accounting for electrode variability across subjects[J]. Advances in Neural Information Processing Systems, 2024, 37: 108600-108624.
>
> [4] Wu D, Li S, Feng C, et al. Towards Homogeneous Lexical Tone Decoding from Heterogeneous Intracranial Recordings[J]. arXiv preprint arXiv:2410.12866, 2024.
>
> [5] Angrick M, Ottenhoff M C, Diener L, et al. Real-time synthesis of imagined speech processes from minimally invasive recordings of neural activity[J]. Communications biology, 2021, 4(1): 1055.
>
> [6] Li C, Wang Y, Wang Y, et al. A Revisit of Total Correlation in Disentangled Variational Auto-Encoder with Partial Disentanglement[J]. arXiv preprint arXiv:2502.02279, 2025.
>
> [7] Singh A, Thomas T, Li J, et al. Transfer learning via distributed brain recordings enables reliable speech decoding[J]. Nature Communications, 2025, 16(1): 8749.

---

### Author Response · Authors · 2025-11-29
**Our Aim and the Evolution of Intracranial Neural Decoding (1/2)**

We sincerely thank the AC and the reviewers for their careful reading and thoughtful feedback. **To further pinpoint the contributions of this work to the field of intracranial decoding in recent years, we have compiled a summary of recent advances in intracranial neural decoding.**

1. Angrick et al. (Communications biology 2021) [1]: **An exploratory study using sEEG for speech decoding.** They recorded intracranial sEEG signals from `1` subject and collected `~500` samples based on an English word reading task. In this work, **researchers were not aware of the importance of channel selection based on functional similarity, resulting in not fully exploiting the potential of the collected data**.

2. BrainBERT (ICLR 2023) [2]: The first work introduces MAE pre-training into intracranial sEEG modeling, constructing a **single-channel-level sEEG foundation model** through representation learning on single-channel sEEG recordings. In this work, they evaluate BrainBERT on a privately collected speech-perception sEEG dataset (`~5.5` hours per subject): they collect sEEG recordings from `10` subjects during watching movies. And BrainBERT outperforms multiple baselines (e.g., Linear, Deep Neural Network).

3. NLA (ICML 2023) [3]: They propose TWM to resolve the temporal misalignment of trials and evaluate TWM on a privately collected vocal-production ECoG dataset: **they collect ECoG recordings from `2` subjects** during speaking pre-defined sentences multiple times, resulting in `~500` and `~1k` samples respectively.

4. Brant (NeurIPS 2023) [4]: **The first work that integrates sEEG recordings from multiple subjects via a temporal-spatial Transformer.** They construct a general sEEG foundation model by modeling relationships among channels. In this work, they evaluate Brant on a privately collected seizure detection sEEG dataset (`~2k` hours across `10` subjects). And Brant outperforms BrainBERT and other advanced models in `4` forecasting tasks and seizure detection.

5. Du-IN (NeurIPS 2024) [5]: They collected intracranial sEEG recordings (`15` hours per subject) from `12` subjects and **released the task recordings to serve as a benchmark**: each subject was instructed to read `61` pre-defined Chinese words for `~50` times, resulting in `~3k` trials per subject. Then, they introduce two-stage pre-training (VQ-VAE+MAE) to advance speech decoding from intracranial sEEG recordings. Furthermore, they **demonstrate the significant disadvantages of previous neurofoundation models [2,4] in decoding complex cognitive states (i.e., `61` word classification), compared to binary discrimination tasks in BrainBERT [2]**.

6. Brain Treebank (NeurIPS 2024) [6]: They **released the task recordings (`~5.5` hours per subject) used in BrainBERT [2] to serve as a benchmark**: each subject watched multiple movies, resulting in multiple binary speech-perception tasks (i.e., pitch, volume, sentence onset detection, and speech/non-speech).

7. PopT (ICLR 2025) [7]: They reframe the `4` binary decoding tasks to ensure balanced positive/negative samples. Then, they **build a spatial Transformer based on channel-wise BrainBERT embeddings to serve as a neurofoundation model for neural decoding**. PopT is pre-trained on `10` subjects, and fine-tuned using subject-specific samples.

Current intracranial neural decoding research primarily follows two paradigms: (1) **subject-specific models** [1,3,5], which are both pre-trained and fine-tuned using subject-specific data, and (2) **neurofoundation models** [2,4,7], which are pre-trained across subjects but **still require subject-specific fine-tuning**. Our work is positioned within the first paradigm.

Our experiments establish that even with data from a single subject, channel selection via our method significantly boosts the performance of subject-specific models trained from scratch (Figure 3a). **By leveraging channel selection based on extracted functional groups (Figure 3), our approach enables a subject-specific model to surpass the performance of multi-subject pre-trained foundation models, regardless of their use of channel selection (Tables 2 & 3).** This result highlights the crucial importance of **effective channel selection** for decoding. Building on this, we can further improve decoding performance through neural disentanglement. In summary, **BrainStratify offers a robust and interpretable solution for decoding speech from intracranial recordings, advancing toward clinically viable and transparent neuroprosthetic systems**.

---

> ### Author Response · Authors · 2025-11-29
> **Our Aim and the Evolution of Intracranial Neural Decoding (2/2)**
>
> **As for cross-subject transfer, recent work [8] attempted to evaluate the model performance under this setting (i.e., trained on `N-1` subjects and tested on the remaining `1` subject), and the model only achieved twice the performance of random models.** Based on our previous exploration in this setting, we believe that, due to differences in participants' brains, the higher the spatial resolution of the recording and the more complex the cognitive task being decoded, the worse cross-subject transfer performance tends to be. The ability to perform more challenging neural decoding tasks using intracranial data enables a greater appreciation of the clinical value of intracranial neural signals. Therefore, **real-world clinical intracranial BCIs typically require subject-specific fine-tuning to achieve optimal performance**. In this work, we mainly explore neural decoding models under a subject-specific fine-tuning setting in order to achieve optimal performance for decoding cognitive tasks, especially speech decoding.
>
> **We sincerely thank the AC and the reviewers again for their careful reading and thoughtful feedback.**
>
> **References**:
>
> [1] Angrick M, Ottenhoff M C, Diener L, et al. Real-time synthesis of imagined speech processes from minimally invasive recordings of neural activity[J]. Communications biology, 2021, 4(1): 1055.
>
> [2] Wang C, Subramaniam V, Yaari A U, et al. BrainBERT: Self-supervised representation learning for intracranial recordings[J]. arXiv preprint arXiv:2302.14367, 2023.
>
> [3] Cho C J, Chang E, Anumanchipalli G. Neural latent aligner: cross-trial alignment for learning representations of complex, naturalistic neural data[C]//International Conference on Machine Learning. PMLR, 2023: 5661-5676.
>
> [4] Zhang D, Yuan Z, Yang Y, et al. Brant: Foundation model for intracranial neural signal[J]. Advances in Neural Information Processing Systems, 2023, 36: 26304-26321.
>
> [5] Zheng H, Wang H, Jiang W, et al. Du-IN: Discrete units-guided mask modeling for decoding speech from Intracranial Neural signals[J]. Advances in Neural Information Processing Systems, 2024, 37: 79996-80033.
>
> [6] Wang C, Yaari A, Singh A, et al. Brain treebank: Large-scale intracranial recordings from naturalistic language stimuli[J]. Advances in Neural Information Processing Systems, 2024, 37: 96505-96540.
>
> [7] Chau G, Wang C, Talukder S, et al. Population transformer: Learning population-level representations of neural activity[J]. ArXiv, 2025: arXiv: 2406.03044 v4.
>
> [8] Wu D, Bu L, Jia Y, et al. Towards Unified Neural Decoding with Brain Functional Network Modeling[J]. bioRxiv, 2025: 2025.06. 02.657011.

---

### Note · Authors · 2026-01-28

I have read and agree with the venue's withdrawal policy on behalf of myself and my co-authors.

---

### Meta-Review · Area_Chair_isvv · 2026-01-07

**Summary:**

The paper a two-stage coarse-to-fine framework for speech decoding from intracranial signals through disentanglement. Six datasets are used for evaluation.

Strength: The reviewers mention the following strengths.

(1) The paper is well written with details.

(2) The proposed method has interesting points.

(3) The analysis of learned codex and articulatory actions is interesting.

(4) Statistical analysis is well performed.

Weakness: (1) The novelty is incremental, and the work is more on the engineering side (Reviewer 5Gy2).

(2) The new datasets contain only one subject, and most modeling is subject-specific. Thus, cross-patient or cross-session performance is not clear (Reviewers 5Gy2 & xusM).

(3) It is unclear which components bring actual improvement, certain design choices are not fully justified (Reviewer 5Gy2).

(4) The title says the paper is about speech decoding, but the framework looks rather general and doesn't seem to be designed specifically for speech decoding (Reviewer 721d). To be a general framework, the paper lacks experimental evidence (Reviewer xusM).

(5) Channel selection (Brainstra-coarse) doesn't bring performance improvement (Reviewers cnbL & 721d).

(6) There is no comparison with supervised deep learning baselines (Reviewer cnbL).

(7) More theoretical/experimental analysis is needed to support the authors' claim (Reviewer xusM).

(8) Experimental comparison is not entirely fair (Reviewer xusM).

(9) How to determine the number of codex groups is unclear (Reviewer xusM).

(10) There is no experimental comparison with the previous studies on disentanglement (Reviewer xusM).

**Reviewer Concerns:**

(1) The authors' rebuttal mentions that the proposed method has a value as an integrated framework, particularly using the spatial attention graphs. AC thinks that the unified framework
is valuable, but as Reviewer 5Gy2 mentioned, innovation beyond integration of known building blocks is rather weak.

(2) The authors' rebuttal mentions the rareness of patients with ALS and common practices in related studies with only 1-2 subjects. It also mentions that subject-wise fine-tuning would be anyway necessary in real applications. The former seems to make sense, while the latter does not so much because the proposed model is not trained with the subject-specific fine-tuning setting.

(3) Appendix K presents ablation studies on channel cluster and DPQ, which seems to answer only partly to the reviewer's comments.

(4) The rebuttal mentions that the framework is for decoding related to cognitive functions, but due to lack of datasets, the evaluation is mostly on speech decoding. AC thinks it is still confusing if the title mentions speech decoding but only evaluation (not the method) is relevant to speech decoding, which blurs the focus of the paper.

(5) The rebuttal presents new results to show BrainStratify-Coarse reduces the number of supervised samples. To AC, this seems to be only an indirect evidence of effectiveness of BrainStratify-Coarse.

(6) The rebuttal provides new results for comparison to simple deep learning baselines, addressing the concern.

(7) The rebuttal provides explanation and small new results. This would resolve the reviewer's concern.

(8) The rebuttal reports new results of existing methods, PopT trained on single subject data, and DUET and CCM with rereferenced data, as in the proposed method. The concern would be resolved.

(9) The rebuttal says determining the number of codex groups is difficult and not the focus of the paper. This doesn't seem to fully resolve the reviewer's concern.

(10) The rebuttal explains why comparison is not feasible. Not sure the concern would be resolved.

**Reviewer Scores:**

Reviewer 5Gy2: AC doesn't think s/he would have changed.

Reviewer cnbL: responded that s/he will increase the score.

Reviewer 721d: AC doesn't think s/he would have changed.

Reviewer xusM: AC doesn't think s/he would have changed.

---

### Decision · Program_Chairs · 2026-01-26

Reject